# On symmetry-resolved generalized entropies

Fei Yan[1], Sara Murciano[2,3], Pasquale Calabrese[4,5] and Robert Konik[1]

**1** Condensed Matter Physics and Materials Science Division,
Brookhaven National Laboratory, Upton, NY 11973, USA
**2** Department of Physics and Institute for Quantum Information and Matter,
California Institute of Technology, Pasadena, CA 91125, USA
**3** Walter Burke Institute for Theoretical Physics,
California Institute of Technology, Pasadena, CA 91125, USA
**4** SISSA and INFN Sezione di Trieste, via Bonomea 265, 34136 Trieste, Italy
**5** International Centre for Theoretical Physics (ICTP),
Strada Costiera 11, 34151 Trieste, Italy

## Abstract

**Symmetry-resolved entanglement, capturing the refined structure of quantum entanglement in systems with global symmetries, has attracted a lot of attention recently. In this manuscript, introducing the notion of symmetry-resolved generalized entropies, we aim to develop a computational framework suitable for the study of excited state symmetry-resolved entanglement as well as the dynamical evolution of symmetry-resolved entanglement in symmetry-preserving out-of-equilibrium settings. We illustrate our framework using the example of (1+1)-d free massless compact boson theory, and benchmark our results using lattice computation in the XX chain. As a byproduct, our computational framework also provides access to the probability distribution of the symmetry charge contained within a subsystem and the corresponding full counting statistics.**

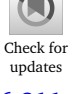

# 1   Introduction

The relevance of entanglement as a tool to understand several physical phenomena and discover new ones is well established nowadays. For instance, entanglement can be used to detect phase transitions [1], to understand out-of-equilibrium problems [2], to study the topological properties of a system [3,4], and to unravel the black hole information paradox [5]. In the presence of global symmetries, the structure of quantum entanglement becomes even more intriguing and an active area of research has been lately dedicated to studying this quantity in the presence of global symmetries [6–8]. This goes under the name of *symmetry resolution of quantum entanglement*, and it has been studied in several contexts, ranging from lattice systems [9–16], to conformal field theories (CFTs) [17–38]. Additionally, recent work of [39–42] have generalized ground state symmetry resolution to non-invertible symmetries. The first input towards the study of symmetry-resolved entanglement was an experiment done by Lukin et al. [43] about the time evolution of a many-body localized system through particle fluctuations and correlations. This work uncovered the relevance of the internal symmetry structure of the total entanglement entropy, which can be better understood in terms of two quantities into which the total entanglement can be split, i.e. the configurational entropy and the number entropy. In the presence of both disorder and interactions, the dynamics of configurational and number entanglement occur over different time scales: the number entanglement quickly saturates to an asymptotic value while the configurational one exhibits a slow logarithmic growth.

After this experiment, symmetry resolution of quantum entanglement has been studied in a plethora of setups. An important topic is the dependence of symmetry-resolved quantum entanglement on symmetry representations. For example, in CFTs with a $U(1)$ symmetry free of 't Hooft anomaly, at leading order in the subsystem size, if we focus on small fluctuations of the subsystem charge around its mean value, symmetry-resolved entanglement does not depend on the charge sector and its leading order behavior coincides with that of the total entanglement entropy [8]. By resolving with respect to the irreducible representations of a non-abelian group, this result also holds for CFTs with non-abelian global symmetries, and the entanglement equipartition is broken only by a constant term depending on the logarithm of the dimension of the irreducible representation [25, 44]. Understanding how quantum entanglement splits into different symmetry representation sectors is worthwhile because we can focus on a specific sector whose Hilbert space dimension is much smaller than the total Hilbert space dimension, thereby reducing the memory resources normally needed in numerical methods. For quantum entanglement of excited states in critical systems or in lattice systems, equipartition can be broken by terms that explicitly depend on the microscopic details of the system or on the parameter that introduces the energy gap [10, 45]. The general lesson is that, as far as the microscopic details of the models are negligible and we focus on small deviations of the charge (i.e. the charge density is not fixed [15, 46]), equipartition of quantum entanglement would hold.

As a byproduct, studying the symmetry resolution also allows us to compute the probability distribution of an observable restricted in a subsystem and the corresponding full counting statistics [47]. A prominent example is the probability distribution for the symmetry charge contained within a subsystem, where the study of symmetry resolution gives us access to the average together with all the moments (in addition to the variance) of the distribution.

Given the rapid developments in this field, our work in this manuscript is motivated from two perspectives, namely the study of symmetry-resolved entanglement for generic excited states, and the dynamical evolution of symmetry-resolved entanglement in a symmetry-preserving out-of-equilibrium setting. In the following, we will briefly describe these two motivations, before giving a summary of our results.

## 1.1 Symmetry-resolved entanglement measures for excited states

Compared with the ground state symmetry-resolved entanglement, relatively less studied are the symmetry-resolved entanglement measures for excited states, which possess distinct properties from the ground state symmetry resolution. For example, Ref. [18] studied symmetry-resolved entropies for low-lying primary excited states in 2d CFTs with a $U(1)$ global symmetry. Focusing on the example of the free compact boson CFT (or Luttinger liquid), they found that entanglement equipartition *still* holds at the leading order in subsystem size, but there exist subleading *universal* terms which break the equipartition. Additional studies of excited state symmetry resolution include Refs. [48–50] which investigated the symmetry resolution due to exciting a finite number of quasiparticles above the ground state of a free integrable quantum field theory.

In addition to the excited state symmetry resolution, symmetry-resolved relative entropies between low-lying excited states in CFTs have also been studied in [20, 27]. However, it remains a challenging task to analyze the symmetry resolution of various entanglement measures beyond the lowest energy states corresponding to primary fields in the CFT. Part of our motivation for this paper is to develop a systematic approach to compute the symmetry-resolved entanglement measures for arbitrary excited states, using as a prototypical example the free massless compact boson theory relevant to the experimental setups involving Luttinger liquids - for an overview see [51–53]. Luttinger liquid physics can include spin-charge separation in 1D metals and nanotubes [54, 55], power-law correlations of the dynamic structure func-

tion in 1D cold atomic systems [56–60], and the fractionalization of magnons into spinons in quasi-1D spin chains [61, 62]. Additionally, the same computational framework can also be applied to the analytical study of the subsystem charge distribution for excited states, with the potential to compare with simulations and experiments on full counting statistics.

## 1.2 Dynamical evolution and generalized entropies

The out-of-equilibrium dynamical evolution of the entanglement and Rényi entropies can reveal the dynamical properties of quantum systems, in addition to their own interesting features. In general it is difficult to track the dynamical evolution of entanglement measures analytically. Suppose that we choose an appropriate basis $\{|\psi_i\rangle\}$ for the Hilbert space of the total closed system equipped with a bipartition $A \cup A^c$, a pure state $|\psi(t)\rangle$ and its corresponding reduced density matrix $\rho_A(t)$ can then be expressed in terms of this basis as follows:

$$|\psi(t)\rangle = \sum_i \alpha_i(t)|\psi_i\rangle, \qquad \rho_A(t) = \mathrm{Tr}_{A^c}|\psi(t)\rangle\langle\psi(t)| = \sum_{i,j} \alpha_i(t)\alpha_j^*(t)\mathrm{Tr}_{A^c}|\psi_i\rangle\langle\psi_j|. \quad (1)$$

It is then clear that, in order to compute entanglement measures built out of $\rho_A(t)$, we have to take into account "cross-term" contributions where the in-state and the out-state are distinct from each other.

This setup has been studied for the standard Rényi entropies in [63], under the notion of *generalized Rényi entanglement entropies*, defined as

$$S_n(\psi_1, \ldots, \psi_{2n}) := \frac{1}{1-n}\log\frac{\mathrm{Tr}_A\left[\mathrm{Tr}_{A^c}(|\psi_{2n-1}\rangle\langle\psi_{2n}|)\ldots\mathrm{Tr}_{A^c}(|\psi_1\rangle\langle\psi_2|)\right]}{\mathrm{Tr}_A\left[\mathrm{Tr}_{A^c}(|0\rangle\langle0|)^n\right]}, \quad (2)$$

where $|\psi_{2i-1}\rangle$ and $\langle\psi_{2i}|$ denote the in-state and the out-state in the $i$-th ($1 \leq i \leq n$) replica copy of the system, and $|0\rangle$ denotes the ground state. In particular, when $\psi_{1,\ldots,2n} = \psi$, Eq. (2) computes the difference between the $n$-th Rényi entropy for the state $|\psi\rangle$ and the ground state $n$-th Rényi entropy, which has been the subject of intensive investigations [64–78]. Additionally, when $|\psi_{2i-1}\rangle = |\psi_1\rangle$ and $\langle\psi_{2i}| = \langle\psi_2|$ for all $i$, Eq. (2) reduces to the so-called pseudo-entropies studied in [79, 80]. By choosing $|\psi_i\rangle$ to be basis states for CFTs and combining with truncated conformal space approach [81–83], the generalized entropies behave as building blocks of a new computational scheme for the dynamical evolution of entanglement measures in out-of-equilibrium (1+1)-d theories. As a testbed of this idea, Ref. [84] applied this framework to compute the time-dependence of the second Rényi entropy for an experimentally-realizable quench protocol [58–60] of joining two Luttinger liquids in their ground state by turning on a coupling between the two copies.

The computational scheme combining the analytical results of generalized Rényi entropies with numerical methods tracking time evolution of the state of the system,[1] represents the only available method to quantify the post-quench dynamics of Rényi entropies in both integrable and chaotic systems, where an intuitive picture of entanglement spreading via quasi-particle excitations is missing. In this work, we aim to generalize this computational scheme to study the entanglement dynamics in symmetry-preserving out-of-equilibrium processes.

## 1.3 Summary of our results

In this work, focusing on quantum systems with a global symmetry $G$, we develop a computational framework for the study of entanglement dynamics and subsystem charge distribution in

---

[1] See Section 2.1 and Ref. [63] for a more detailed description of such a computational scheme. The numerical method applied in this framework is Truncated Conformal Space Approach (TCSA), which is the only reliable numerical method which allows direct study of out-of-equilibrium quantum field theories. We remark that a promising alternative method to study non-equilibrium quantum field theories has been explored in [85], which applied the time-dependent variational principle to the variational manifold of continuous matrix product states.

symmetry-preserving out-of-equilibrium settings. Although this framework can be generalized to generic symmetry groups, here we will focus on the case of $G = U(1)$.

We introduce the notion of *symmetry-resolved generalized Rényi entropies* capturing quantum entanglement within a fixed subsystem charge sector, defined as

$$S_n(q; \psi_1, ..., \psi_{2n}) = \frac{1}{1-n} \log \frac{\text{Tr}_A \left[ \text{Tr}_{A^c} (|\psi_{2n-1}\rangle\langle\psi_{2n}|) ... \text{Tr}_{A^c} (|\psi_1\rangle\langle\psi_2|) \Pi_q \right]}{\text{Tr}_A \left[ \text{Tr}_{A^c}^n |0\rangle\langle 0| \Pi_q \right]}, \tag{3}$$

where $\Pi_q$ denotes the projector into the sector where the subsystem $A$ contains $U(1)$ charge $q \in \mathbb{Z}$. As a special application, setting $\psi_i = \psi$ ($i = 1, ..., 2n$) gives us access to the symmetry-resolved Rényi entropies for an arbitrary excited state $|\psi\rangle$.

To facilitate the computation of symmetry-resolved generalized entropies $S_n(q; \psi_1, ..., \psi_{2n})$, we introduce the *normalized generalized charged moments*

$$F_n(\theta; \psi_1, \ldots, \psi_{2n}) := \frac{\text{Tr}_A \left[ \text{Tr}_{A^c} (|\psi_{2n-1}\rangle\langle\psi_{2n}|) \ldots \text{Tr}_{A^c} (|\psi_1\rangle\langle\psi_2|) e^{i\theta Q_A} \right]}{\text{Tr}_A \left[ \text{Tr}_{A^c}^n |0\rangle\langle 0| e^{i\theta Q_A} \right]}, \tag{4}$$

where $\theta \in (-\pi, \pi]$ represents an Aharonov-Bohm flux coupled to the subsystem-restricted $U(1)$ charge operator $Q_A$. This quantity can be viewed as a generalization of the ground state charged moment introduced in [6], to accommodate the dynamical evolution in symmetry-preserving out-of-equilibrium settings. In addition to their roles in computing the symmetry-resolved generalized entropies, the normalized generalized charged moments have their own physical meaning. In particular, $F_1(\theta; \psi_1, ..., \psi_{2n})$ can be viewed as the generating function for the generalized subsystem charge distribution, behaving as building blocks in the computation of full counting statistics.

Motivated by experimental interest in systems represented by Luttinger liquids, we demonstrate our computational framework in the context of a free massless compact boson theory. We outline the replica-trick computation of normalized generalized charged moments $F_n(\theta; \psi_1, ..., \psi_{2n})$, providing explicit summation formulae for the case of $n = 1$ in Eq. (58) and for the case of $n = 2$ in Eq. (81). We benchmark our results using lattice computation in the XX chain. Given knowledge of the generalized charged moments, we outline the computation of generalized subsystem charge distribution and symmetry-resolved generalized second Rényi entropy, where we observe universal subleading terms which break the entanglement equipartition.

This paper is organized as follows. In Section 2 we define symmetry-resolved generalized entropies and give a generic outline of their computation in 2d CFTs. In Section 3 we set up our conventions in the compact boson CFT and describe the computation of the generalized charged moments. We then give explicit summation formulae of the $n = 1$ and the $n = 2$ generalized charged moments in Section 4 and Section 5 respectively, and benchmark our results using lattice computation in the XX chain. Finally, we outline the computation of the generalized subsystem charge distribution in Section 6 and the computation of the symmetry-resolved generalized second Rényi entropy in Section 7, before concluding an outlook on this work in Section 8.

## 2 Generalities of the symmetry-resolved generalized entropies

In this Section, we will introduce the notion of symmetry-resolved generalized entropies and outline their computations. In Section 2.1, we will illustrate the definitions and ideas for theories with a tensor-product Hilbert space, such as 1d spin chains. We then move on to outline the computations in the context of 2d conformal field theories in Section 2.2.

## 2.1 Motivation and definition of symmetry-resolved generalized entropies

We consider a (1+1)-d system equipped with a bipartition of the spacial dimension. Throughout this subsection, we work under the assumption that the bipartition is "clear-cut", namely the total Hilbert space admits a tensor-product decomposition $\mathcal{H} = \mathcal{H}_A \otimes \mathcal{H}_{A^c}$. Examples where this applies includes (1+1)-d quantum spin chains where the physical degrees of freedom reside on the qubit sites and the subsystem $A$ consists of a subset of qubits.

Our setup in this paper concerns closed systems, where the total system is in a pure state described by $\rho_{\text{tot}} = |\psi\rangle\langle\psi|$. Tracing out the complement then gives the reduced density matrix for the subsystem $A$,

$$\rho_A := \text{Tr}_{A^c}\rho_{\text{tot}} = \text{Tr}_{A^c}|\psi\rangle\langle\psi|. \tag{5}$$

Quantum entanglement between the subsystem $A$ and its complement can be captured by the entanglement entropy (von Neumann entropy) defined as

$$S_{\text{vN}} := -\text{Tr}_A[\rho_A\log\rho_A]. \tag{6}$$

For systems with a global symmetry $G$, quantum entanglement can be refined according to different representations under the global symmetry. Given a symmetry operator $U_g$ supported on the whole space implementing the symmetry action labeled by $g \in G$, under certain conditions one can define a symmetry operator $U_{g,A}$ which only acts on the subsystem $A$. This construction is possible if the operator $U_{g,A}$ ends topologically on a chosen symmetry-preserving entanglement cut boundary condition imposed at the boundaries of $A$ [22,25,86]. For spin chains with on-site symmetries, the canonical entanglement cut boundary condition is the free/open boundary condition, therefore $U_{g,A}$ is well-defined as long as $U_g$ is compatible with the open boundary condition. Suppose that the subsystem-restricted symmetry operators $U_{g,A}$ are well-defined, then for any state of the total system $\rho_{\text{tot}}$ satisfying $[\rho_{\text{tot}}, U_g] = 0$ for all $g \in G$, its corresponding reduced density matrix would satisfy $[\rho_A, U_{g,A}] = 0$ for all $g \in G$. This follows from

$$U_{g,A}\rho_A U_{g,A}^\dagger = \text{Tr}_{A^c}\left(U_{g,A}\rho_{\text{tot}}U_{g,A}^\dagger\right) = \text{Tr}_{A^c}\left(U_{g^{-1},A^c}U_g\rho_{\text{tot}}U_g^\dagger U_{g^{-1},A^c}^\dagger\right) = \rho_A. \tag{7}$$

As a result, the reduced density matrix can be written as a direct sum over different super-selection sectors labeled by representations of $G$. Within each sector, we can define and compute the entanglement entropy and Rényi entropies as per usual.

In this paper we will focus on the case of $G = U(1)$. The symmetry operator corresponding to a $U(1)$ group element $g = e^{i\theta}$ is given by $U_g = e^{i\theta Q}$, where $Q$ represents the $U(1)$ charge operator. The corresponding subsystem-restricted symmetry operator is $U_{g,A} = e^{i\theta Q_A}$, with $Q_A$ measuring the $U(1)$ charge for the subsystem $A$. The reduced density matrix $\rho_A$ can then be written as a direct sum over super-selection sectors labeled by $q \in \mathbb{Z}$ representing the $U(1)$ charge contained in the subsystem $A$. Concretely we have

$$\rho_A = \bigoplus_{q\in\mathbb{Z}}\rho_A\Pi_q, \tag{8}$$

where $\Pi_q$ denotes the projector into the charge-$q$ sector, explicitly given by

$$\Pi_q = \int_{-\pi}^{\pi}\frac{d\theta}{2\pi}e^{-i\theta q}e^{i\theta Q_A}. \tag{9}$$

The entanglement entropy restricted to the charge-$q$ sector is defined using the normalized charge-$q$ sector density matrix as follows:

$$S_{\text{vN}}(q) = -\text{Tr}_A[\rho_{A,q}\log\rho_{A,q}], \qquad \rho_{A,q} := \frac{\rho_A\Pi_q}{\text{Tr}_A[\rho_A\Pi_q]}. \tag{10}$$

The symmetry-resolved $n$-th Rényi entropy is defined via $\rho_{A,q}$ as

$$\mathcal{S}_n(q) := \frac{1}{1-n}\log\text{Tr}_A\left[\rho_{A,q}^n\right] = \frac{1}{1-n}\log\frac{\text{Tr}_A\left[\rho_A^n\Pi_q\right]}{\text{Tr}_A^n\left[\rho_A\Pi_q\right]}, \tag{11}$$

where the $n \to 1$ limit produces the symmetry-resolved entanglement entropy $\mathcal{S}_{\text{vN}}(q)$.

As described in the Introduction, the symmetry-resolved generalized entanglement and Rényi entropies can be motivated by an analytical framework suitable for computing the dynamical evolution of these quantities in symmetry-preserving out-of-equilibrium settings of the closed total system. Given a basis $\{|\psi_i\rangle\}$ of the total Hilbert space, at any time $t$ the total system is in a pure state given by

$$|\psi(t)\rangle = \sum_i \alpha_i(t)|\psi_i\rangle. \tag{12}$$

For symmetry-preserving non-equilibrium settings where the initial state $|\psi(0)\rangle$ is a symmetry-eigenstate carrying a particular charge, the $|\psi_i\rangle$ appearing on the right-hand side of (12) are basis states for the corresponding symmetry charge sector. The time-dependent total density matrix is then given by

$$\rho_{\text{tot}}(t) = \sum_{i,j} \alpha_i(t)\alpha_j^*(t)\rho_{ij}, \qquad \rho_{ij} := |\psi_i\rangle\langle\psi_j|. \tag{13}$$

Correspondingly, the time-dependent reduced density matrix is

$$\rho_A(t) = \sum_{i,j} \alpha_i(t)\alpha_j^*(t)\rho_{ij,A}, \qquad \rho_{ij,A} = \text{Tr}_{A^c}\rho_{ij}. \tag{14}$$

Here $\rho_{ij,A}$ can be regarded as a generalized reduced density matrix satisfying $\text{Tr}_A\rho_{ij,A} = \delta_{ij}$. Moreover, as the non-equilibrium setting is symmetry-preserving and $\psi_{i,j}$ carry the same symmetry charge, $\rho_{ij,A}$ commutes with the symmetry action restricted to the subsystem $A$.

The time-dependent symmetry-resolved entanglement entropy and Rényi entropies can be computed in terms of $\rho_{ij,A}$ together with the projector into the desired charge sector. In this paper we aim to develop a framework suitable for computing the time-dependence of symmetry-resolved $n$-th Rényi entropy, which is obtained by substituting Eq. (14) into Eq. (11). Concretely, we have

$$\mathcal{S}_n(q,t) = \frac{1}{1-n}\log\frac{\displaystyle\sum_{i_1,...,i_{2n}} \alpha_{i_1}(t)\alpha_{i_2}^*(t)...\alpha_{i_{2n-1}}(t)\alpha_{i_{2n}}^*(t)\text{Tr}_A\left[\rho_{i_{2n-1}i_{2n},A}...\rho_{i_1i_2,A}\Pi_q\right]}{\displaystyle\sum_{j_1,...,j_{2n}} \alpha_{j_1}(t)\alpha_{j_2}^*(t)...\alpha_{j_{2n-1}}(t)\alpha_{j_{2n}}^*(t)\text{Tr}_A\left[\rho_{j_1j_2,A}\Pi_q\right]...\text{Tr}_A\left[\rho_{j_{2n-1}j_{2n},A}\Pi_q\right]}. \tag{15}$$

We would like to identify fundamental quantities for the evaluation of $\mathcal{S}_n(q,t)$ in Eq. (15). Motivated from the global quench setup where the total system is in its ground state before the quench, we consider relative quantities with respect to the ground state observables. Denoting

$$G_n(q;\psi_1,...,\psi_{2n}) := \frac{\text{Tr}_A\left[\rho_{(2n-1)(2n),A}...\rho_{12,A}\Pi_q\right]}{\text{Tr}_A\left[\text{Tr}_{A^c}^n|0\rangle\langle 0|\Pi_q\right]}, \tag{16}$$

we define the *symmetry-resolved generalized $n$-th Rényi entropy* as

$$\begin{aligned} S_n(q;\psi_1,...,\psi_{2n}) &:= \frac{1}{1-n}\log G_n(q;\psi_1,...,\psi_{2n}) \\ &= \frac{1}{1-n}\log\frac{\text{Tr}_A\left[\text{Tr}_{A^c}\left(|\psi_{2n-1}\rangle\langle\psi_{2n}|\right)...\text{Tr}_{A^c}\left(|\psi_1\rangle\langle\psi_2|\right)\Pi_q\right]}{\text{Tr}_A\left[\text{Tr}_{A^c}^n|0\rangle\langle 0|\Pi_q\right]}, \end{aligned} \tag{17}$$

where $|\psi_{2i-1}\rangle\langle\psi_{2i}|$ ($i = 1, \ldots, n$) denotes the generalized density matrix in the $i$-th replica copy. A physically meaningful post-quench observable is $\Delta\mathcal{S}_n(q, t) = \mathcal{S}_n(q, t) - \mathcal{S}_n(q, 0)$, namely the change of symmetry-resolved $n$-th Rényi entropy with respect to its pre-quench value. For cases where the total system is in its ground state before the quench, from Eq. (15) we see that $\Delta\mathcal{S}_n(q, t)$ can then be computed using the knowledge of $G_n(q; \psi_{i_1}, \ldots, \psi_{i_{2n}})$ and $G_1(q; \psi_{i_1}, \psi_{i_2})$. This framework is also applicable to quench setups where the total system is in a certain excited state before the quench, where the post-quench change of symmetry-resolved $n$-th Rényi entropy is obtained by further subtracting the difference between the excited state and ground state symmetry-resolved $n$-th Rényi entropy.

One of our goals in this paper is to obtain analytical expressions of the symmetry-resolved generalized $n$-th Rényi entropy in Eq. (17) for 2d conformal field theories. We will outline the general methodology in Section 2.2, before demonstrating the procedure in the example of the free compact boson CFT.

## 2.2 Computation in 2d conformal field theories

Here we outline the computation strategy for the symmetry-resolved generalized $n$-th Rényi entropy in 2d CFTs. We consider a 2d CFT placed on a cylinder whose circumference is $L$, where the spacial dimension is the circular direction of the cylinder parameterized by $x \sim x + L$. The subsystem $A$ is given by a single interval $[u, v]$ on the spacial circle. The characteristic length scale in this cylindrical geometry is the *chord length* defined as

$$\ell := \frac{L}{\pi}\sin(\pi r), \qquad r = \frac{v - u}{L}. \tag{18}$$

An important strategy in computing the symmetry-resolved entropies and their generalized version is to perform the computation in "Fourier space". By a Fourier transformation, symmetry-resolved entropies are related to the charged moments first introduced in [6, 8], which can be computed by the appropriate CFT partition functions with the insertion of the subsystem-restricted symmetry operator $U_{g,A}$. In a similar fashion, for the computation of symmetry-resolved generalized entropies, we introduce the *normalized $n$-th generalized charged moment* $F_n(\theta; \psi_1, \ldots, \psi_{2n})$, defined as

$$F_n(\theta; \psi_1, \ldots, \psi_{2n}) := \frac{\text{Tr}_A\left[\text{Tr}_{A^c}(|\psi_{2n-1}\rangle\langle\psi_{2n}|)\ldots\text{Tr}_{A^c}(|\psi_1\rangle\langle\psi_2|)e^{i\theta Q_A}\right]}{\text{Tr}_A\left[\text{Tr}_{A^c}^n|0\rangle\langle0|e^{i\theta Q_A}\right]}, \tag{19}$$

where the normalization in the denominator is given by the ground state $n$-th charged moment. We remark that the $n = 1$ generalized charged moment also has its own physical meaning as the generating function for the subsystem charge distribution. The symmetry-resolved generalized $n$-th Rényi entropy can then be computed as

$$S_n(q; \psi_1, \ldots, \psi_{2n}) = \frac{1}{1-n}\log\frac{\int_{-\pi}^{\pi}\frac{d\theta}{2\pi}e^{-i\theta q}F_n(\theta; \psi_1, \ldots, \psi_{2n})\text{Tr}_A\left[\text{Tr}_{A^c}^n|0\rangle\langle0|e^{i\theta Q_A}\right]}{\int_{-\pi}^{\pi}\frac{d\theta}{2\pi}e^{-i\theta q}\text{Tr}_A\left[\text{Tr}_{A^c}^n|0\rangle\langle0|e^{i\theta Q_A}\right]}. \tag{20}$$

Before delving into the CFT computation, we briefly remark on certain regularization issues. In the context of continuum field theories, the Hilbert space of the total system does not admit a tensor-product decomposition anymore, one needs to regularize the entangling points at $x = u$ and $x = v$ by cutting out a small disk with radius $\epsilon$ and placing a chosen boundary condition $a$ at the boundary of the disk [87, 88]. In the setup to study symmetry-resolved entropies, the entanglement cut boundary condition $a$ has to be chosen such that the subsystem-restricted symmetry operators $U_{g,A}$ can end topologically on $a$ [22, 25, 86]. The benefit of our setup is that the observables we consider are defined with respect to the ground state observables, such that there is little dependence on the choice of the boundary condition.

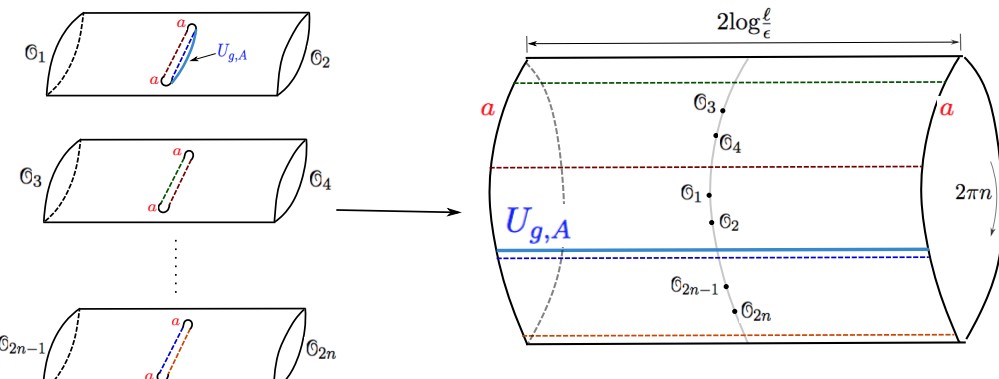

Figure 1: *Left:* The replica geometry $C_n$ for the computation of the generalized charged moment, as a $n$-fold branched covering of a cylinder with infinite length. The $n$ sheets are glued together by identifying the edges of cut-open slits with the same color, e.g. the maroon-colored slit edge on sheet 1 is identified with the maroon-colored slit edge on sheet 2 etc. The operator $\mathbb{O}_{2i-1}$ corresponding to $|\psi_{2i-1}\rangle$ are inserted at the past infinity of the cylinder while $\mathbb{O}_{2i}$ corresponding to $\langle\psi_{2i}|$ are inserted at the future infinity of the cylinder. The subsystem-restricted symmetry operator $U_{g,A}$ is inserted on sheet 1 spanning between the entanglement cut boundaries with boundary condition $a$. *Right:* After a conformal transformation, the replica geometry $C_n$ can be mapped to an annulus with length $2\log\frac{\ell}{\epsilon}$ and circumference $2\pi n$. The operators $\mathbb{O}_{2i-1}$ and $\mathbb{O}_{2i}$ $(i = 1, ..., n)$ are inserted along the median of the annulus.

The computation of generalized charged moment $F_n(\theta; \psi_1, ..., \psi_{2n})$ requires the evaluation of the CFT partition function on a $n$-fold branched covering of the cylinder denoted as $C_n$, with the insertion of a subsystem-restricted symmetry operator $U_{g,A}$ spanning between the entanglement cut boundaries, together with additional insertions of $\mathbb{O}_{2i-1}$ and $\mathbb{O}_{2i}$ $(i = 1, ..., n)$ at the past and future infinities of the cylindrical geometry respectively. This is illustrated on the left-hand side (LHS) of Figure 1. Via a conformal transformation, we are lead to compute a twisted annulus partition function, where the operators $\mathbb{O}_{2i-1}$ and $\mathbb{O}_{2i}$ are inserted along the median of the annulus, as illustrated on the RHS of Figure 1. In the special case where all the operators $\mathbb{O}_{2i-1}$ and $\mathbb{O}_{2i}$ are identity operators, this setup in the compact boson CFT (with compactification radius $1/\beta$) yields the known result for the ground state $n$-th charged moment of the $U(1)$ symmetry resolution [25]

$$\mathrm{Tr}_A\big[\mathrm{Tr}_{A^c}^n|0\rangle\langle 0|e^{i\theta Q_A}\big] = \frac{|\langle a_\theta|\theta\rangle|^2}{|\langle a|0\rangle|^{2n}} e^{\frac{1}{6}\left(\frac{1}{n}-n\right)\log\frac{\ell}{\epsilon}} e^{-\frac{\beta^2\theta^2}{2\pi^2 n}\log\frac{\ell}{\epsilon}} , \qquad (21)$$

where $|a_\theta\rangle$ denotes the $U(1)$ symmetric entanglement cut boundary state, and $|\theta\rangle$ is the ground state in the Hilbert space on $S^1$ with a $U(1)$-twist by $e^{i\theta}$. Performing the Fourier transform then yields:

$$\int_{-\pi}^{\pi} \frac{d\theta}{2\pi} e^{-i\theta q} \mathrm{Tr}_A\big[\mathrm{Tr}_{A^c}^n|0\rangle\langle 0|e^{i\theta Q_A}\big] = \int_{-\pi}^{\pi} \frac{d\theta}{2\pi} e^{-i\theta q} \frac{|\langle a_\theta|\theta\rangle|^2}{|\langle a|0\rangle|^{2n}} e^{\frac{1}{6}\left(\frac{1}{n}-n\right)\log\frac{\ell}{\epsilon}} e^{-\frac{\beta^2\theta^2}{2\pi^2 n}\log\frac{\ell}{\epsilon}}$$

$$\sim |\langle a|0\rangle|^{2-2n} e^{\frac{1}{6}\left(\frac{1}{n}-n\right)\log\frac{\ell}{\epsilon}} \sqrt{\frac{n\pi}{2\beta^2\log\frac{\ell}{\epsilon}}} e^{-\frac{n\pi^2 q^2}{2\beta^2\log\frac{\ell}{\epsilon}}} , \qquad (22)$$

where in the second line we have expanded the integration domain to $\theta \in (-\infty, \infty)$ and approximated the twisted g-function $|\langle a_\theta|\theta\rangle|$ using the untwisted one. The Fourier transform behaves as a Gaussian, where the variance is proportional to $\frac{\beta^2}{n}\log\frac{\ell}{\epsilon}$.

For generic CFTs, it is difficult to obtain an analytical expression for the generalized charged moment $F_n(\theta;\psi_1,...,\psi_{2n})$. In this paper, we will focus on the free compact boson theory where the calculation is manageable. Details on this are spelled out in Section 3, Section 4, and Section 5. Concretely, the normalized generalized charged moment $F_n(\theta;\psi_1,\dots,\psi_{2n})$ is approximately given by a polynomial in $\theta$, with finite coefficients being trigonometric functions in the subsystem size ratio $r$.

Our main goal is to extract physical quantities at the leading orders in the chord length $\ell$. With this in mind, we can approximately evaluate the symmetry-resolved generalized Rényi entropy in Eq. (20) by performing the Fourier transformation in a similar fashion as in Eq. (22). The final result will be expressed in terms of the chord length $\ell$, while the entanglement cut boundary condition dependence cancels out between the numerator and denominator of Eq. (20). We perform this calculation for the generalized subsystem charge distribution in Section 6 and for the symmetry-resolved generalized second Rényi entropy in Section 7.

# 3 The case of free compact boson theory

Even though the computation frame that we develop here applies to a generic 2d CFT, we will demonstrate our methodology using the prototypical example of the massless free compact boson. Consider the massless free compact boson theory put on the cylinder with circumference $L$. We denote the compactification radius as $1/\beta$, namely

$$\phi \sim \phi + \frac{2\pi}{\beta}\,. \tag{23}$$

Our convention for the Euclidean action is

$$\mathcal{L}(\phi) = \frac{1}{8\pi} \int_0^L dx\,d\tau\, \partial^\mu \phi(x,\tau) \partial_\mu \phi(x,\tau), \tag{24}$$

and we name the dual compact boson $\tilde{\phi}(x,\tau)$. Using the complex coordinate $w = x + i\tau$ on the cylinder, we have[2]

$$\phi(w,\bar{w}) = \phi_L(w) + \phi_R(\bar{w}), \qquad \tilde{\phi}(w,\bar{w}) = \phi_L(w) - \phi_R(\bar{w}). \tag{25}$$

The mode expansion of $\phi(w,\bar{w})$ is given by

$$\phi(w,\bar{w}) = \phi_0 - \frac{2\pi}{L}\pi_0(w - \bar{w}) - \frac{\pi m}{\beta L}(w + \bar{w}) + i\sum_{k \neq 0} \frac{1}{k}\left(a_k e^{\frac{2\pi ik}{L}w} - \bar{a}_{-k} e^{\frac{2\pi ik}{L}\bar{w}}\right), \tag{26}$$

where $\phi_0$ is the bosonic zero mode, $\pi_0$ is its conjugate momentum which takes values in $\{n\beta | n \in \mathbb{Z}\}$, and $m \in \mathbb{Z}$ is the winding number. The creation modes are:

$$
\begin{aligned}
a_{-k} &= -\frac{1}{2\pi} e^{-\frac{2\pi k}{L}\tau} \int_0^L dx\, e^{\frac{2\pi ik}{L}x} \partial_w \phi\,, \\
\bar{a}_{-k} &= \frac{1}{2\pi} e^{-\frac{2\pi k}{L}\tau} \int_0^L dx\, e^{-\frac{2\pi ik}{L}x} \partial_{\bar{w}} \phi\,.
\end{aligned}
\tag{27}
$$

Generic states in the Hilbert space on the spacial circle are obtained by acting with the left- and right-moving creation modes on the highest weight states:

$$|\psi\rangle = \mathcal{N} \prod_{j=1}^{N_L} a_{-k_j} \prod_{\bar{j}=1}^{N_R} \bar{a}_{-\tilde{k}_{\bar{j}}} |n,m\rangle, \tag{28}$$

---

[2]In our convention the left-moving part is holomorphic in the cylinder coordinate $w = x + i\tau$, which corresponds to the "right-moving" part in Minkowski signature.

where the highest weight states $|n, m\rangle$ ($n, m \in \mathbb{Z}$) are obtained by acting with the corresponding vertex operators on the vacuum

$$|n, m\rangle = \lim_{\tau \to -\infty} V_{n\beta + \frac{m}{2\beta}, n\beta - \frac{m}{2\beta}}(w, \bar{w})|0\rangle \, . \tag{29}$$

Our convention for vertex operators on the cylinder is

$$V_{\alpha, \bar{\alpha}}(w, \bar{w}) = \left(\frac{\mathrm{i}L}{2\pi} e^{\frac{2\pi \mathrm{i}}{L} w}\right)^{\frac{\alpha^2}{2}} \left(-\frac{\mathrm{i}L}{2\pi} e^{-\frac{2\pi \mathrm{i}}{L} \bar{w}}\right)^{\frac{\bar{\alpha}^2}{2}} : e^{\mathrm{i}\alpha \phi_L(w) + \mathrm{i}\bar{\alpha} \phi_R(\bar{w})} : \, , \tag{30}$$

such that after the conformal transformation to the complex plane coordinated by $e^{-\frac{2\pi \mathrm{i}}{L} w}$, one recovers the usual definition of vertex operator on the plane. The factor $\mathcal{N}$ in Eq. (28) ensures that the state $|\psi\rangle$ is properly normalized. Concretely, given the list $\{k_j, j = 1, ..., N_L\}$ ($\{\tilde{k}_{\tilde{j}}, \tilde{j} = 1, ..., N_R\}$) labeling the chiral (anti-chiral) creation modes, we count each distinct entry $k_\ell$ ($\tilde{k}_{\tilde{\ell}}$) with its multiplicity $n_\ell$ ($\tilde{n}_{\tilde{\ell}}$), then the normalization is given by

$$\mathcal{N} = \prod_\ell \frac{1}{\left(\sqrt{k_\ell}\right)^{n_\ell} \sqrt{n_\ell !}} \prod_{\tilde{\ell}} \frac{1}{\left(\sqrt{\tilde{k}_{\tilde{\ell}}}\right)^{\tilde{n}_{\tilde{\ell}}} \sqrt{\tilde{n}_{\tilde{\ell}} !}} \, . \tag{31}$$

In the following, we use a short-hand notation for the state $|\psi\rangle$ in Eq. (28) as

$$|\psi\rangle := |k_1, \ldots, k_{N_L}; \tilde{k}_1, \ldots, \tilde{k}_{N_R}; \alpha, \bar{\alpha}\rangle \, , \tag{32}$$

where $\alpha = n\beta + m/(2\beta)$ and $\bar{\alpha} = n\beta - m/(2\beta)$.

The free compact boson theory has two $U(1)$ symmetries: the $U(1)_m$ momentum symmetry and the $U(1)_w$ winding symmetry. Even though there is a mixed anomaly between these two symmetries, either $U(1)$ symmetry is by itself free of 't Hooft anomaly, and we can study the corresponding $U(1)$-symmetry resolution of entanglement entropy. Motivated by the global quench setup preserving the $U(1)_w$ symmetry, here we will focus on the $U(1)_w$ symmetry resolution. The $U(1)_m$ symmetry resolution of entanglement entropy works out in an analogous manner.

We can write down the subsystem-restricted charge operator $Q_A$ explicitly for each of the $U(1)$ symmetries in terms of the symmetry twist operators inserted at the entanglement-cut boundaries. The subsystem in our setup is a single interval $A = [u, v]$ on the spatial circle. After a regularization with a cutoff of size $\epsilon$, the endpoint coordinates are denoted as $w_0 = u + \epsilon$ and $w_{0'} = v - \epsilon$. The charges supported along $A_\epsilon := [w_0, w_{0'}]$ are given by

$$\begin{aligned} U(1)_m \, &: \, Q_{A,\mathrm{m}} = -\frac{1}{4\pi\beta} \int_{A_\epsilon} dx\, \partial_x \tilde{\phi} \, , \\ U(1)_w \, &: \, Q_{A,\mathrm{w}} = -\frac{\beta}{2\pi} \int_{A_\epsilon} dx\, \partial_x \phi \, . \end{aligned} \tag{33}$$

Therefore in the compact boson theory, the symmetry generated by $e^{\mathrm{i}\theta Q_A}$ can be implemented explicitly by inserting two vertex operators at $w_0$ and $w_{0'}$ on one sheet of the replica geometry. Using the conventions of Eq. (30), the twist operators implementing the $U(1)$ momentum symmetry and the $U(1)$ winding symmetry can be realized, respectively, by the insertions of:

$$\begin{aligned} U(1)_m \, &: \, e^{\mathrm{i}\theta Q_{A,m}} = \left(\frac{2\pi}{L}\right)^{\frac{\theta^2}{4\pi^2\beta^2}} V_{\frac{\theta}{4\pi\beta}, -\frac{\theta}{4\pi\beta}}(w_0, \bar{w}_0) V_{-\frac{\theta}{4\pi\beta}, \frac{\theta}{4\pi\beta}}(w_{0'}, \bar{w}_{0'}) \, , \\ U(1)_w \, &: \, e^{\mathrm{i}\theta Q_{A,w}} = \left(\frac{2\pi}{L}\right)^{\frac{\theta^2\beta^2}{2\pi^2}} V_{\frac{\theta\beta}{2\pi}, \frac{\theta\beta}{2\pi}}(w_0, \bar{w}_0) V_{-\frac{\theta\beta}{2\pi}, -\frac{\theta\beta}{2\pi}}(w_{0'}, \bar{w}_{0'}) \, . \end{aligned} \tag{34}$$

The regularization at each endpoint of $A$ is obtained by cutting off a small disk of radius $\epsilon$, while imposing boundary conditions at the boundary of the disk (entangling surface) [87,88]. To investigate the symmetry resolution of the entanglement, one needs to choose *symmetry-preserving* boundary conditions [22, 25, 86] at the entangling surface. In our setup, the resolution of the momentum symmetry $U(1)_m$ requires Neumann boundary conditions on $\phi$ at the entangling surface, while for the $U(1)_w$ winding symmetry resolution, we need to impose Dirichlet boundary conditions on the field $\phi$.

## 3.1 The normalized generalized charged moment in free compact boson theory

Here we describe the computation of the normalized n-th generalized charged moment introduced in Section 2.2 in the free compact boson theory. Using the convention in Eq. (32), we denote the states as

$$|\psi_i\rangle = |k_{i,1}, \ldots, k_{i,N_{L_i}}; \tilde{k}_{i,1}, \ldots, \tilde{k}_{i,N_{R_i}}; \alpha_i, \bar{\alpha}_i\rangle, \qquad i = 1, \ldots, 2n. \tag{35}$$

Recall that the normalized n-th generalized charged moment $F_n(\theta; \psi_1, \ldots, \psi_{2n})$ is defined as:

$$F_n(\theta; \psi_1, \ldots, \psi_{2n}) := \frac{\text{Tr}_A\left[\text{Tr}_{A^c}(|\psi_{2n-1}\rangle\langle\psi_{2n}|)\ldots\text{Tr}_{A^c}(|\psi_1\rangle\langle\psi_2|e^{i\theta Q_A})\right]}{\text{Tr}_A\left(\text{Tr}_{A^c}^n|0\rangle\langle 0|e^{i\theta Q_A}\right)}, \tag{36}$$

where $|\psi_{2i-1}\rangle\langle\psi_{2i}|$ $(i = 1, \ldots, n)$ denotes the generalized density matrix built from the states (32) on the $i$-th sheet, while the normalization in the denominator is given by the ground state $n$-th charged moment. We make a few comments about some properties of $F_n(\theta; \psi_1, \ldots, \psi_{2n})$, which will be useful later:

- The chiral and anti-chiral contributions to $F_n(\theta; \psi_1, \ldots, \psi_{2n})$ factorize.

- The difference between the $U(1)_m$ momentum symmetry and the $U(1)_w$ winding symmetry is reflected in the different expressions for $e^{i\theta Q_A}$ in Eq. (34). Due to the factorization between chiral and anti-chiral contributions, there are only minor differences between the $U(1)_m$ and $U(1)_w$ symmetry resolution results. In this manuscript, we will focus only on the $U(1)_w$ symmetry resolution.

- After the normalization given by the ground state charged moment, the normalized generalized charged moment is a finite function in $\theta$. As we will see later, at leading order $F_n(\theta; \psi_1, \ldots, \psi_{2n})$ is a polynomial in $\theta$ whose coefficients are trigonometric functions in the subsystem size ratio $r$. The regularization-dependent corrections are power-law suppressed by $\log\frac{\ell}{\epsilon}$, where $\ell$ is the *chord length* defined in (18). For large $\ell/\epsilon$, the contributions to $F_n(\theta; \psi_1, \ldots, \psi_{2n})$ from the sub-leading corrections are negligible. We will therefore focus on computing $F_n(\theta; \psi_1, \ldots, \psi_{2n})$ as a polynomial in $\theta$.

We are now ready to write down the explicit expression of $F_n(\theta; \psi_1, \ldots, \psi_{2n})$ for the $U(1)_w$ winding symmetry. Due to the factorization between the chiral and anti-chiral part, we can write

$$\begin{aligned} F_n(\theta; \psi_1, \ldots, \psi_{2n}) = {} & F_n^{\text{L}}(\theta; \{k_{1,1}, \ldots, k_{1,N_{L_1}}\}, \ldots, \{k_{2n,1}, \ldots, k_{2n,N_{L_{2n}}}\}; \{\alpha_i\}) \\ & \times F_n^{\text{R}}(\theta; \{\tilde{k}_{1,1}, \ldots, \tilde{k}_{1,N_{R_1}}\}, \ldots, \{\tilde{k}_{2n,1}, \ldots, \tilde{k}_{2n,N_{R_{2n}}}\}; \{\bar{\alpha}_i\}). \end{aligned} \tag{37}$$

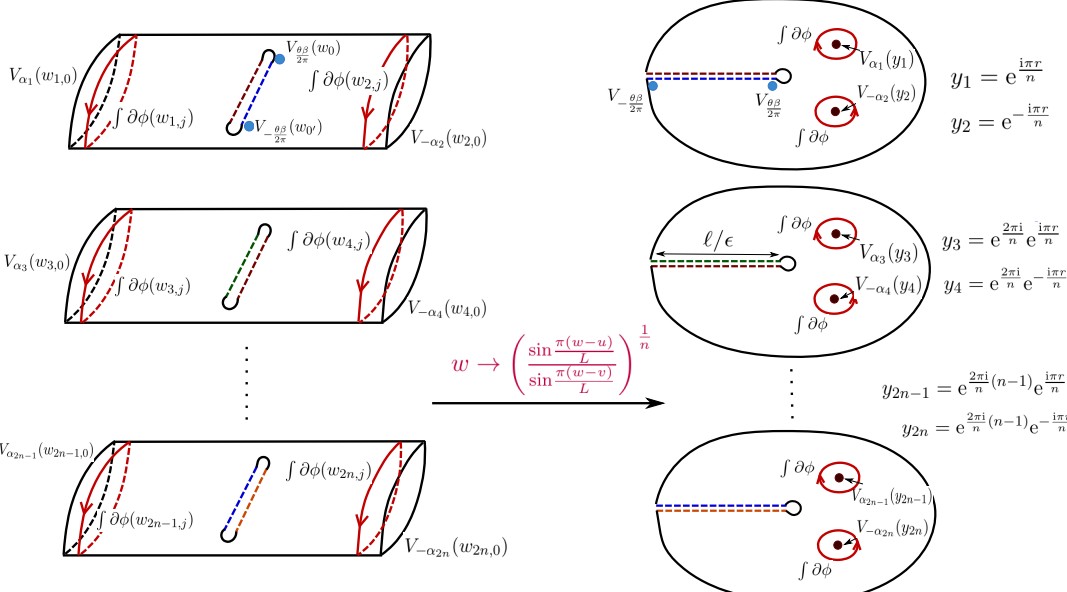

Figure 2: *Left:* The replica geometry $C_n$ for computing the chiral normalized charged moment, as a $n$-fold branched covering of a cylinder with infinite length. The $n$ sheets are glued together by identifying the edges of cut-open slits with the same color, e.g. the maroon-colored slit edge on sheet 1 is identified with the maroon-colored slit edge on sheet 2 etc. The vertex operators $V_{\sigma_i \alpha_i}$ and $\partial \phi$ are inserted at the past and future infinities of the cylinder. The symmetry-twist vertex operators $V_{\pm \frac{\theta \beta}{2\pi}}$ are inserted on sheet 1 by the regularized branch points. *Right:* After the conformal transformation in Eq. (40), the replica geometry $C_n$ is mapped to a $n$-fold covering of the plane with cut-out disks at 0 and $\infty$.

Using Eq. (27), the left-moving contribution is given by

$$F_n^{\mathrm{L}}(\theta; \{k_{1,1}, \ldots, k_{1,N_{\mathrm{L}_1}}\}, \ldots, \{k_{2n,1}, \ldots, k_{2n,N_{\mathrm{L}_{2n}}}\}; \{\alpha_i\}) = \prod_{i=1}^{2n} \mathcal{N}_i \left(\frac{-1}{2\pi}\right)^{N_{\mathrm{L}_i}} \prod_{j=1}^{N_{\mathrm{L}_i}} e^{-\frac{2\pi}{L}\sigma_i k_{i,j} \tau_{i,j}}$$

$$\times \int_0^L dx_{i,j} e^{\frac{2\pi \mathrm{i}}{L}\sigma_i k_{i,j} x_{i,j}} \frac{\left\langle V_{\frac{\theta\beta}{2\pi}}(w_0) V_{-\frac{\theta\beta}{2\pi}}(w_{0'}) \prod_{i=1}^{2n} V_{\sigma_i\alpha_i}(w_{i,0}) \prod_{j=1}^{N_{\mathrm{L}_i}} \partial\phi(w_{i,j}) \right\rangle_{C_n}}{\left\langle V_{\frac{\theta\beta}{2\pi}}(w_0) V_{-\frac{\theta\beta}{2\pi}}(w_{0'}) \right\rangle_{C_n}}. \quad (38)$$

Here $\mathcal{N}_i$ is the left-moving part of the normalization factor given in Eq. (31), $\sigma_{2i-1} = 1$ and $\sigma_{2i} = -1$ for $i = 1, \ldots n$ distinguish between the in-states and the out-states in Eq. (19). The vertex operators $V_{\sigma_i\alpha_i}$ are inserted at $w_{i,0} := x_{i,0} + \mathrm{i}\tau_{i,0}$ and $\partial\phi$ are inserted at $w_{i,j>0} := x_{i,j} + \mathrm{i}\tau_{i,j}$. For odd (even) $i$, we have $\tau_{i,j} \to -\infty$ ($\tau_{i,j} \to +\infty$) respectively. Each vertex operator $V_{\sigma_i\alpha_i}$ comes from the (chiral) highest weight part of the state (32), while the field $\partial\phi$ comes from the oscillator contribution. The correlation functions are evaluated on the replica geometry $C_n$, as a $n$-fold branched covering of the cylinder. This is shown on the LHS of Figure 2, where we also demonstrate the locations of different field insertions. Without loss of generality, the symmetry twist operators $V_{\frac{\theta\beta}{2\pi}}$ and $V_{-\frac{\theta\beta}{2\pi}}$ are inserted at the entanglement cut boundaries on the 1st sheet.

Similarly, the right-moving contribution is

$$F_n^{\mathrm{R}}(\theta;\{\tilde{k}_{1,1},\ldots,\tilde{k}_{1,N_{\mathrm{R}_1}}\},\ldots,\{\tilde{k}_{2n,1},\ldots,\tilde{k}_{2n,N_{\mathrm{R}_{2n}}}\};\{\bar{\alpha}_i\}) = \prod_{i=1}^{2n}\widetilde{\mathcal{N}}_i\left(\frac{1}{2\pi}\right)^{N_{\mathrm{R}_i}}\prod_{j=1}^{N_{\mathrm{R}_i}}e^{-\frac{2\pi}{L}\sigma_i\tilde{k}_{i,j}\tilde{\tau}_{i,j}}$$

$$\times\int_0^L d\tilde{x}_{i,j}e^{-\frac{2\pi\mathrm{i}}{L}\sigma_i\tilde{k}_{i,j}\tilde{x}_{i,j}}\frac{\left\langle V_{\frac{\theta\beta}{2\pi}}(\bar{w}_0)V_{-\frac{\theta\beta}{2\pi}}(\bar{w}_{0'})\prod_{i=1}^{2n}V_{\sigma_i\bar{\alpha}_i}(\bar{w}_{i,0})\prod_{j=1}^{N_{\mathrm{R}_i}}\bar{\partial}\phi(\overline{\tilde{w}}_{i,j})\right\rangle_{C_n}}{\left\langle V_{\frac{\theta\beta}{2\pi}}(\bar{w}_0)V_{-\frac{\theta\beta}{2\pi}}(\bar{w}_{0'})\right\rangle_{C_n}}, \quad (39)$$

where the notation here is similar to that of Eq. (38).

To compute the left-/right-moving normalized generalized charged moment, it is convenient to perform the following conformal transformation

$$w \longrightarrow z = \left(\frac{\sin\frac{\pi(w-u)}{L}}{\sin\frac{\pi(w-v)}{L}}\right)^{\frac{1}{n}}. \quad (40)$$

This maps the replica geometry $C_n$ to a $n$-fold branched covering of the plane with boundaries at $z=0$ and $z=\infty$, where the branch cuts are located along the negative real axis. This is illustrated on the RHS of Figure 2. We remark that this conformal map composed with the logarithmic map yields the conformal transformation illustrated in Figure 1. By exploiting this mapping, in Section 4 and Section 5, we will explicitly compute the normalized charged moments for the cases $n=1$ and $n=2$.

# 4 The $n=1$ generalized charged moment

In this Section, we compute the $n=1$ generalized charged moment, which can be regarded as the generating function for the probability distribution of the symmetry charge contained in the subsystem $A$. For example, the expectation value of the subsystem charge is given by

$$\langle Q_A\rangle = \mathrm{Tr}_A[Q_A\rho_A] = -\mathrm{i}\frac{\partial}{\partial\theta}\mathrm{Tr}_A\left[\rho_A e^{\mathrm{i}\theta Q_A}\right]\Big|_{\theta=0}. \quad (41)$$

Similarly, by taking higher derivatives in $\theta$ we obtain higher moments of the subsystem charge distribution. If the total system is in a $U(1)$ symmetric ground state, then the subsystem charge distribution is a Gaussian distribution with $\langle Q_A\rangle = 0$. If the total system is in a generic excited state, then $\langle Q_A\rangle$ is equal to the $U(1)$ charge of the state for the whole system rescaled by the subsystem size ratio, as the charge density on average is spatially uniform.

In the following, we will explicitly compute the generalized $n=1$ charged moment in the example of the free massless compact boson theory.

## 4.1 A sum formula for the $n=1$ generalized charged moment

The normalized $n=1$ generalized charged moment is given by

$$F_1(\theta;\psi_1,\psi_2) = \frac{\mathrm{Tr}_A\left(\mathrm{Tr}_{A^c}|\psi_1\rangle\langle\psi_2|e^{\mathrm{i}\theta Q_A}\right)}{\mathrm{Tr}_A\left(\mathrm{Tr}_{A^c}|0\rangle\langle0|e^{\mathrm{i}\theta Q_A}\right)} = F_1^{\mathrm{L}}(\theta;\psi_1,\psi_2)F_1^{\mathrm{R}}(\theta;\psi_1,\psi_2). \quad (42)$$

As the right-moving contribution is related to the left-moving contribution by complex conjugation, in the following we will focus on the left-moving part.

Figure 3: *Left:* The cylindrical geometry $C_1$ for computing the chiral $n = 1$ normalized generalized charged moment, together with locations of various field insertions. *Right*: After the conformal transformation in Eq. (44), the cylindrical geometry $C_1$ is mapped to the complex place with two disks cut out around 0 and $\infty$, which we denote as $\mathbb{C}'$.

Using the convention in Eq. (32) for labeling the states, the left-moving/chiral part of the normalized $n = 1$ generalized charged moment is

$$
\begin{aligned}
F_1^{\mathrm{L}}(\theta; \psi_1, \psi_2) &= F_1^{\mathrm{L}}(\theta; \{k_{1,1}, \ldots, k_{1,N_{\mathrm{L}_1}}\}, \{k_{2,1}, \ldots, k_{2,N_{\mathrm{L}_2}}\}; \{\alpha_1, \alpha_2\}) \\
&= \prod_{i=1}^{2} \mathcal{N}_i \left(-\frac{1}{2\pi}\right)^{N_{\mathrm{L}_i}} \prod_{j=1}^{N_{\mathrm{L}_i}} e^{-\frac{2\pi}{L}\sigma_i k_{i,j}\tau_{i,j}} \int_0^L dx_{i,j} e^{\frac{2\pi i}{L}\sigma_i k_{i,j} x_{i,j}} \\
&\quad \times \frac{\left\langle V_{\frac{\theta\beta}{2\pi}}(w_0) V_{-\frac{\theta\beta}{2\pi}}(w_{0'}) V_{\alpha_1}(w_{1,0}) V_{-\alpha_2}(w_{2,0}) \prod_{j=1}^{N_{\mathrm{L}_1}} \partial\phi(w_{1,j}) \prod_{j=1}^{N_{\mathrm{L}_2}} \partial\phi(w_{2,j}) \right\rangle_{C_1}}{\left\langle V_{\frac{\theta\beta}{2\pi}}(w_0) V_{-\frac{\theta\beta}{2\pi}}(w_{0'}) \right\rangle_{C_1}},
\end{aligned}
\tag{43}
$$

where $C_1$ is a cylinder parameterized by $w = x + i\tau$. The normalization factors $\mathcal{N}_i$ are given by the chiral part of Eq. (31). The symmetry twist operators are inserted on the real axis at $w_0 = u + \epsilon$ and $w_{0'} = v - \epsilon$. The vertex operators and $\partial\phi$ associated with $|\psi_1\rangle$ and $|\psi_2\rangle$ are inserted at $w_{1,j\geq0} = x_{1,j} - i\infty$ and $w_{2,j\geq0} = x_{2,j} + i\infty$ respectively. We illustrate the cylindrical geometry $C_1$ together with the various field insertions on the LHS of Figure 3.

To evaluate the chiral normalized $n = 1$ generalized charged moment, we perform a conformal map from the cylinder to the plane:

$$
w \to z = \frac{\sin\frac{\pi}{L}(w-u)}{\sin\frac{\pi}{L}(w-v)}.
\tag{44}
$$

In particular, the vertex operator insertion locations $w_{1,0}$ and $w_{2,0}$ are mapped respectively to

$$
y_1 = e^{i\pi r}, \qquad y_2 = e^{-i\pi r}.
\tag{45}
$$

Moreover, integrals of the form $\int_0^L dx_{1,j}\partial\phi(w_{1,j})$ $(\int_0^L dx_{2,j}\partial\phi(w_{2,j}))$ are mapped to clockwise (counterclockwise) contour integrals around $y_1$ ($y_2$). Pictorially this is illustrated on the RHS of Figure 3. Additionally, we have

$$
e^{\frac{2\pi i}{L}x} = e^{\frac{2\pi}{L}\tau}e^{\frac{2\pi i}{L}v}\frac{z-e^{-i\pi r}}{z-e^{i\pi r}}.
\tag{46}
$$

The chiral normalized $n = 1$ generalized charged moment in Eq. (43) then becomes

$$F_1^L(\theta; \psi_1, \psi_2) = \prod_{i=1}^{2} \mathcal{N}_i \left(-\frac{1}{2\pi}\right)^{N_{L_i}} e^{\frac{2\pi i}{L} v \sum_{j=1}^{N_{L_i}} \sigma_i k_{i,j}} \prod_{j=1}^{N_{L_i}} \oint_{C_{\sigma_i}^{i,j}} dz_{i,j} \left(\frac{\partial z}{\partial w}\bigg|_{w_{i,0}} \left(\pm \frac{iL}{2\pi} e^{\pm \frac{2\pi i}{L} w_{i,0}}\right)\right)^{\frac{\alpha_i^2}{2}} \quad (47)$$

$$\times \left(\frac{z_{i,j} - y_i^*}{z_{i,j} - y_i}\right)^{k_{i,j}} \frac{\langle V_{\frac{\beta\theta}{2\pi}}(y_0) V_{-\frac{\beta\theta}{2\pi}}(y_{0'}) V_{\alpha_1}(y_1) V_{-\alpha_2}(y_2) \prod_{j=1}^{N_{L_1}} \partial\phi(z_{1,j}) \prod_{j=1}^{N_{L_2}} \partial\phi(z_{2,j})\rangle_{\mathbb{C}'}}{\langle V_{\frac{\beta\theta}{2\pi}}(y_0) V_{-\frac{\beta\theta}{2\pi}}(y_{0'})\rangle_{\mathbb{C}'}},$$

where $\sigma_i$ is as defined in Eq. (38) and $C_{\sigma_i=1}$ ($C_{\sigma_i=-1}$) is a clockwise (counterclockwise) contour around $z_{i,j}$. The correlation functions inside the integrand in Eq. (47) are evaluated on the complex plane with two disks cut out around 0 and $\infty$, which we denote as $\mathbb{C}'$. The locations for various field insertions are illustrated on the RHS of Figure 3. Notice that the vertex operators within the correlator follow the usual convention of vertex operators on the plane, as the prefactors from Eq. (30) have been pulled out in the first line of the above equation.

To explicitly compute $F_1^L(\theta; \psi_1, \psi_2)$ in Eq. (47), we can express the correlation function ratio in the integrand as an explicit sum according to different choices of Wick contractions. As can be seen from Figure 1, the effects of entanglement cut boundary conditions on the correlation function ratio are suppressed by powers of $\log(\ell/\epsilon)$. In this work, we focus on the leading order results, where we approximate the correlation functions on $\mathbb{C}'$ using the complex plane correlation functions. In particular, the two-point functions are given by

$$\langle V_\alpha(y) \partial\phi(z)\rangle_{\mathbb{C}} = \frac{i\alpha}{y - z},$$
$$\langle \partial\phi(z_1) \partial\phi(z_2)\rangle_{\mathbb{C}} = -\frac{1}{(z_1 - z_2)^2}. \quad (48)$$

For notational convenience, we denote $\alpha_0' := \beta\theta/(2\pi)$, $\alpha_1' = \alpha_1$ and $\alpha_2' = -\alpha_2$. The correlation function ratio in Eq. (47) can be written as

$$\frac{\langle V_{\alpha_0'}(y_0) V_{-\alpha_0'}(y_{0'}) \prod_{i=1}^{2} V_{\alpha_i'}(y_i) \prod_{j=1}^{N_{L_1}} \partial\phi(z_{1,j}) \prod_{j=1}^{N_{L_2}} \partial\phi(z_{2,j})\rangle_{\mathbb{C}'}}{\langle V_{\alpha_0'}(y_0) V_{-\alpha_0'}(y_{0'})\rangle_{\mathbb{C}'}} \sim \prod_{i=1}^{2} \langle V_{\alpha_0'}(y_0) V_{\alpha_i'}(y_i)\rangle_{\mathbb{C}} \quad (49)$$

$$\times \prod_{i=1}^{2} \langle V_{-\alpha_0'}(y_{0'}) V_{\alpha_i'}(y_i)\rangle_{\mathbb{C}} \langle V_{\alpha_1'}(y_1) V_{\alpha_2'}(y_2)\rangle_{\mathbb{C}}$$

$$\times \Bigg[ \langle \partial\phi(z_{1,1})...\partial\phi(z_{1,N_{L_1}}) \partial\phi(z_{2,1})...\partial\phi(z_{2,N_{L_2}})\rangle_{\mathbb{C}}$$

$$+ \sum_{(i_1,j_1)} \left(\sum_{k=0,1,2} \langle V_{\alpha_k'}(y_k) \partial\phi(z_{i_1,j_1})\rangle_{\mathbb{C}}\right) \left\langle \prod_{(i,j)\neq(i_1,j_1)} \partial\phi(z_{i,j})\right\rangle_{\mathbb{C}}$$

$$+ \sum_{\substack{(i_1,j_1), \\ (i_2,j_2)}} \left(\sum_{k=0,1,2} \langle V_{\alpha_k'}(y_k) \partial\phi(z_{i_1,j_1})\rangle_{\mathbb{C}}\right) \left(\sum_{k=0,1,2} \langle V_{\alpha_k'}(y_k) \partial\phi(z_{i_2,j_2})\rangle_{\mathbb{C}}\right) \left\langle \prod_{\substack{(i,j)\neq(i_1,j_1), \\ (i_2,j_2)}} \partial\phi(z_{i,j})\right\rangle_{\mathbb{C}}$$

$$+ \cdots \prod_{\substack{i=1,2 \\ j=1,...,N_{L_i}}} \left(\sum_{k=0,1,2} \langle V_{\alpha_k'}(y_k) \partial\phi(z_{i,j})\rangle_{\mathbb{C}}\right) \Bigg].$$

From the correlator ratio expansion in Eq. (49), it is clear that to obtain a non-zero result, we have to impose the condition $\alpha_1 = \alpha_2 = \alpha$, from now on we will only keep the $\alpha$-dependence.

We now use Eq. (49) to write the chiral normalized $n = 1$ generalized charged moment as a sum of contributions organized according to the Wick contractions:

$$F_1^L(\theta; \psi_1, \psi_2) = \left(\prod_{i=1}^2 \mathcal{N}_i\right)\left(-\frac{1}{2\pi}\right)^{N_{L_1}+N_{L_2}} \xi(\{k_{i,j}\}) K(\alpha) \sum_{\substack{\{k_{m,n}\}, \\ \{k_{p,q}\}}} g_{V\partial\phi}(\theta, \alpha, \{k_{m,n}\}) g_{\partial\phi\partial\phi}(\{k_{p,q}\}), \quad (50)$$

where:

- The normalization factors $\mathcal{N}_i$ are as given in Eq. (31). The phase factor $\xi$ is given by

$$\xi(\{k_{i,j}\}) = \exp\left[\frac{2\pi i}{L} \nu \sum_{i=1}^2 \sum_{j=1}^{N_{L_1}} \sigma_i k_{i,j}\right]. \quad (51)$$

  When restricting to the zero-momentum sector, the phases $\xi$ coming from the left- and right-moving contributions will cancel out.

- The prefactor $K(\alpha)$ contains contributions purely from the vertex operators, which correspond to the primary state part in $|\psi_{1,2}\rangle$.

$$\begin{aligned}
K(\alpha) &= \prod_{i=1}^2 \langle V_{\frac{\beta\theta}{2\pi}}(y_0) V_{\alpha_i'}(y_i)\rangle \prod_{i=1}^2 \langle V_{-\frac{\beta\theta}{2\pi}}(y_{0'}) V_{\alpha_i'}(y_i)\rangle \langle V_\alpha(y_1) V_{-\alpha}(y_2)\rangle \\
&\quad \times \prod_{i=1}^2 \left(\frac{\partial z}{\partial w}\bigg|_{w_{i,0}} \left(\pm\frac{iL}{2\pi} e^{\pm\frac{2\pi i}{L} w_{i,0}}\right)\right)^{\frac{\alpha^2}{2}} \\
&= y_1^{\frac{\beta\theta}{\pi}\alpha}(y_1 - y_2)^{-\alpha^2}(2i\sin(\pi r))^{\alpha^2} = e^{i\beta\theta r\alpha}.
\end{aligned} \quad (52)$$

- The summation in Eq. (50) is organized according to different Wick contraction choices for the correlator ratio. We denote locations for the $\partial\phi$ insertions as $z_{i,j}$, where the first index $i \in \{1,2\}$ labels the pole $y_i$ that the contour integral $\oint dz_{i,j}\partial\phi(z_{i,j})$ circles around. The contour integrals $\oint dz_{i,j}\partial\phi(z_{i,j})$ originate from the creation modes $a_{-k_{i,j}} (i = 1,2; j = 1,...,N_{L_i})$ acting on the primary state in $|\psi_i\rangle$. Given the total collection of $\{k_{i,j}\}$, we enumerate all possible ways to divide this set into two subsets $\{k_{m,n}\}$ and $\{k_{p,q}\}$. For the first subset $\{k_{m,n}\}$, the corresponding $\partial\phi(z_{m,n})$ are contracted with vertex operators. While for the second subset $\{k_{p,q}\}$, the corresponding $\partial\phi(z_{p,q})$ are contracted between themselves.

- The function $g_{V\partial\phi}(\theta, \alpha, \{k_{m,n}\})$ encodes contributions from Wick contractions of $\partial\phi$ with the vertex operators. Concretely we have

$$g_{V\partial\phi}(\theta, \alpha, \{k_{m,n}\}) = \prod_{(m,n)} \sigma_m \beta\theta\left(e^{-\sigma_m 2\pi i r k_{m,n}} - 1\right). \quad (53)$$

  The product is performed over all elements in the set $\{k_{m,n}\}$. Notice that the dependence on the vertex operator charge $\alpha$ drops out in the final answer. We refer to Appendix A.1 for the detailed derivation of Eq. (53).

- The function $g_{\partial\phi\partial\phi}(\{k_{p,q}\})$ encodes contributions from Wick contractions of $\partial\phi$ with themselves. In particular, $g_{\partial\phi\partial\phi}(\{k_{p,q}\})$ is zero unless the set $\{k_{p,q}\}$ contains an even number of elements. Given the collection $\{k_{p,q}\}$ with $2l$ elements, we first enumerate all possibilities of forming $l$ pairs $(k_{p_1,q_1}, k_{r_1,s_1}),...,(k_{p_l,q_l}, k_{r_l,s_l})$. The contribution from each

pair-up possibility is then given by the product over contributions from each individual pair $(k_{p_i,q_i}, k_{r_i,s_i})$. Namely

$$g_{\partial\phi\partial\phi}(\{k_{p,q}\}) = \sum_{\substack{\{(k_{p_1,q_1}, k_{r_1,s_1}) \\ \dots (k_{p_l,q_l}, k_{r_l,s_l})\}}} \prod_{(k_{p_i,q_i}, k_{r_i,s_i})} d(p_i, r_i, k_{p_i,q_i}, k_{r_i,s_i}), \tag{54}$$

where $d(p_i, r_i, k_{p_i,q_i}, k_{r_i,s_i})$ captures the contribution from the two-point function $\langle \partial\phi(z_{p_i,q_i}) \partial\phi(z_{r_i,s_i})\rangle_{\mathbb{C}}$. Concretely it is given by

$$\begin{aligned} d(p, r, k_{p,q}, k_{r,s}) &= 4\pi^2 k_{p,q}, \quad \text{if } p \neq r \text{ and } k_{p,q} = k_{r,s}, \\ &= 0, \qquad\qquad \text{otherwise.} \end{aligned} \tag{55}$$

In other words, contribution from the two-point function $\langle \partial\phi(z_{p,q}) \partial\phi(z_{r,s})\rangle_{\mathbb{C}}$ vanishes, unless the integration contours for $z_{p,q}$ and $z_{r,s}$ circle around different poles, and their integrands has the same pole order. We again refer to Appendix A.1 for a detailed discussion.

We remark on an important property of Eq. (50). As can be seen in Eq. (53), the function $g_{V\partial\phi}(\theta, \alpha, \{k_{m,n}\})$ ends up to be independent of the vertex operator charge $\alpha$. Consequently, the only $\alpha$-dependence for the $n = 1$ normalized generalized charged moment comes from the phase factor $K(\alpha) = e^{i\beta\theta r\alpha}$, multiplied by a similar phase from the right-moving contribution, such that the final result carries an overall phase $e^{i\beta\theta rm}$ where $m$ is the $U(1)_w$ winding charge of the total system, see Eq. (26). This implies that the subsystem charge distribution for a charge-$m$ total state is a shifted distribution, where the expectation value is shifted to the rescaled charge $rm$ ($r$ is the ratio between the subsystem size and the total system size). This is as expected because the average charge density is spatially uniform.

### 4.1.1 The normalized $n = 1$ generalized charged moment as an explicit sum

We can now work out the sum in Eq. (50) as follows. For notational convenience, we denote the total number of left-moving creation modes as $M = N_{L_1} + N_{L_2}$, and there are in total $M$ contour integral variables $z_{1,2,\dots,M}$ with pole orders $k_{1,2,\dots,M}$. The input data to compute the chiral $n = 1$ normalized generalized charged moment consist of

- The vertex operator charge $\alpha$.

- The list of $a_{-k}$ modes acting on the primary states, denoted as $\{k_1, k_2, \dots k_M\}$. This list can be split according to whether the $a_{-k}$ mode acts on the in-state $|\psi_1\rangle$ or the out-state $|\psi_2\rangle$, namely

$$\{k_1, k_2, \dots, k_M\} = \{k_{1,1}, k_{1,2}, \dots, k_{1,N_{L_1}}\}, \{k_{2,1}, k_{2,2}, \dots, k_{2,N_{L_2}}\}. \tag{56}$$

The sum formula for $F_1^L(\theta; \{k_{i,j}\}; \alpha)$ can be conveniently re-written in a recursive fashion. Combining the normalization factors $\mathcal{N}_i$ and the powers of $-1/(2\pi)$ with the function $g_{\partial\phi\partial\phi}(\{k_{p,q}\})$, we define

$$\begin{aligned} R_{k_1,\dots,k_M} &= R_{\{k_{1,1},\dots,k_{1,N_{L_1}}\},\{k_{2,1},\dots,k_{2,N_{L_2}}\}} \\ &= \begin{cases} 1, & \text{if } (k_{1,1},\dots,k_{1,N_{L_1}}) = (k_{2,1},\dots,k_{2,N_{L_2}}) \text{ as unordered lists,} \\ 0, & \text{otherwise.} \end{cases} \end{aligned} \tag{57}$$

Additionally, we denote $n_{k_i}$ as the multiplicity for $k_i$ within its corresponding $k$-list for either the in-state $|\psi_1\rangle$ or the out-state $|\psi_2\rangle$. We further define a variable $\sigma_{k_i}$ such that, if $k_i$ is associated with $|\psi_1\rangle$, then $\sigma_{k_i} = 1$, otherwise $\sigma_{k_i} = -1$. Then we have

$$
\begin{aligned}
F_1^{\mathrm{L}}(\theta; \{k_i\}; \alpha) = \xi(\{k_i\}) e^{i\beta\theta r\alpha} \Bigg( & R_{k_1,\dots,k_M} + \left(-\frac{\theta}{2\pi}\right) \sum_{i=1}^M R_{k_1,\dots,\not{k}_i,\dots,k_M} N(k_i) L(k_i) \\
& + \frac{\theta^2}{4\pi^2} \sum_{i>j=1}^M R_{k_1,\dots,\not{k}_j,\dots,\not{k}_i,\dots,k_M} N(k_i, k_j) L(k_i) L(k_j) \\
& + \cdots + \left(-\frac{\theta}{2\pi}\right)^M N(k_1,\dots,k_M) \prod_{i=1}^M L(k_i) \Bigg).
\end{aligned}
\tag{58}
$$

In the following, we will describe in more details the various ingredients involved in this formula.

- This sum formula recursively uses the quantity in Eq. (57), organizing the result in terms of a polynomial in $\theta$. To clarify the notations, as an example, $R_{k_1,\dots,\not{k}_j,\dots,\not{k}_i,\dots,k_M}$ takes as input a $k$-list obtained by deleting $k_i$ and $k_j$ from the list $\{k_1, k_2, \dots, k_M\}$, and spells out either 1 or 0 according to Eq. (57).

- The function $N(k_i)$ takes care of the normalization factors for the deleted $k$-values in $R_{k_1,\dots,k_M}$. For example, by removing only one mode, we obtain

$$
N(k_i) = \frac{1}{\sqrt{k_i}\sqrt{n_{k_i}}},
\tag{59}
$$

  while if we remove two modes we need to distinguish between two different cases

$$
N(k_i, k_j) = \begin{cases} \frac{1}{k_i \sqrt{n_{k_i}(n_{k_i}-1)}}, & k_i = k_j \text{ and } \sigma_{k_i} = \sigma_{k_j}, \\ \frac{1}{\sqrt{k_i k_j}\sqrt{n_{k_i} n_{k_j}}}, & \text{otherwise}, \end{cases}
\tag{60}
$$

  according to whether the modes are equal and they both belong to the same in-state or out-state.

- The function $L(k_i)$ arises from contribution of the Wick contraction between $\partial\phi(z_i)$ and the vertex operators.

$$
L(k_i) = \sigma_{k_i}\beta\left(e^{-\sigma_{k_i}2\pi i r k_i} - 1\right).
\tag{61}
$$

- Many terms in Eq. (58) will end up vanishing. In particular, if the total number of $a_{-k}$ modes $M$ is even (odd), then the sum reduces to only the terms with an even (odd) power of $\theta$. Additional terms could also vanish if they fail to satisfy the condition in Eq. (57).

## 4.2 Examples and benchmarks of the results

We now provide some more explicit examples of the formula we derived above. In particular, we first compute the generating function of the subsystem charge distribution for arbitrary excited states, we then compute it for the conformal level-2 chiral states.

### 4.2.1 Generating function of the charge distribution for excited states

As an application of the explicit sum formula Eq. (58), here we will work out the $n = 1$ charged moment, which can be viewed as the generating function of the subsystem charge distribution, when the total system is in a generic excited state $|\psi\rangle$. Notice that this is a special case of the $n = 1$ generalized charged moment, where both the in-state $|\psi_1\rangle$ and the out-state $|\psi_2\rangle$ are set to be equal to $|\psi\rangle$, given by

$$|\psi\rangle = \frac{1}{\prod\limits_{i=1}^{l} k_i^{n_i/2}\sqrt{n_i!}}\frac{1}{\prod\limits_{\bar{i}=1}^{\bar{l}} \tilde{k}_{\bar{i}}^{\tilde{n}_{\bar{i}}/2}\sqrt{\tilde{n}_{\bar{i}}!}}\prod_{i=1}^{l}(a_{-k_i})^{n_i}\prod_{\bar{i}=1}^{\bar{l}}(\bar{a}_{-\tilde{k}_{\bar{i}}})^{\tilde{n}_{\bar{i}}}|\alpha,\bar{\alpha}\rangle\,. \tag{62}$$

Due to the factorization of left- and right-moving contributions, we again focus on the left-moving result and Eq. (58) reads

$$e^{-i\beta\theta r\alpha}F_1^{\text{L}}(\theta;\{k_i,n_i\},\{k_i,n_i\};\alpha) \tag{63}$$

$$= 1 - \theta^2\sum_{j=1}^{l}\frac{\sin^2(k_j\pi r)}{\pi^2}\frac{(C_{n_j}^1)^2}{n_j k_j} + \theta^4\left(\sum_{\substack{j,\\ n_j\geq 2}}\frac{\sin^4(k_j\pi r)}{\pi^4}\frac{(C_{n_j}^2)^2}{n_j(n_j-1)k_j^2} + \sum_{\substack{p,q\\ k_p\neq k_q}}\frac{\sin^2(k_p\pi r)\sin^2(k_q\pi r)}{\pi^4}\frac{(C_{n_p}^1 C_{n_q}^1)^2}{n_p n_q k_p k_q}\right)$$

$$+ \cdots + \theta^{2\sum\limits_{j=1}^{l}n_j}\prod_{j=1}^{l}\frac{\sin^{2n_j}(k_j\pi r)}{(-\pi^2)^{n_j}k_j^{n_j}n_j!}\,,$$

where $C_n^m = \binom{m}{n}$ is the number of possibilities to choose $m$ unordered elements from a total of $n$ elements. To benchmark Eq. (63), we can take special limits to recover the charged moments for excited states computed in Ref. [18]. If we start from $|\psi^{\text{L}}\rangle = \frac{1}{\sqrt{k}}a_{-k}|0\rangle$, we find

$$F_1^{\text{L}}(\theta;\{k\},\{k\};0) = 1 - \beta^2\theta^2\frac{\sin^2(k\pi r)}{k\pi^2}\,, \tag{64}$$

which for $k = 1$ matches with the result found in [18] for the (chiral) primary field $i\partial\phi$. In the absence of oscillatory modes, we simply find $F_1^{\text{L}}(\theta;\{0\},\{0\};\alpha) = e^{i\beta\theta r\alpha}$, that also matches with the findings of [18] when the excited state is created by a vertex operator $V_\alpha$.

### 4.2.2 The $n = 1$ generalized charged moments involving level-2 chiral states

As a further example, we compute the charged moments involving conformal level-2 chiral states, i.e. all the combinations involving

$$|\psi_1\rangle = \frac{1}{\sqrt{2}}a_{-1}a_{-1}|\alpha\rangle\,, \qquad |\psi_2\rangle = \frac{1}{\sqrt{2}}a_{-2}|\alpha\rangle\,. \tag{65}$$

As the dependence on the vertex operator charge $\alpha$ is only a phase, in the following we will express our results in the case of $\alpha = 0$.

The chiral normalized $n = 1$ generalized charged moment, for all possible combinations of $|\psi_1\rangle$ and $|\psi_2\rangle$, are given by

$$F_1^{\text{L}}(\theta;\{1,1\},\{1,1\};0) = 1 - \frac{2\beta^2\theta^2}{\pi^2}\sin^2(\pi r) + \frac{\beta^4\theta^4}{2\pi^4}\sin^4(\pi r)\,,$$

$$F_1^{\text{L}}(\theta;\{1,1\},\{2\};0) = F_1^{\text{L}}(\theta;\{2\},\{1,1\};0) = -i\frac{\beta^3\theta^3}{\pi^3}\sin^3(\pi r)\cos(\pi r)\,, \tag{66}$$

$$F_1^{\text{L}}(\theta;\{2\},\{2\};0) = 1 - \frac{\beta^2\theta^2}{2\pi^2}\sin^2(2\pi r)\,.$$

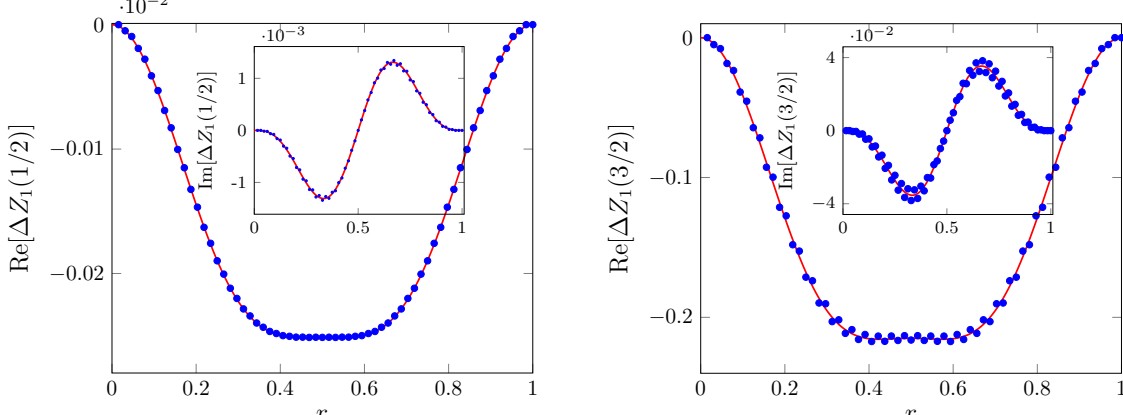

Figure 4: Benchmark of the CFT results for the generalized charged moments in the XX spin chain of size $L = 64$, with periodic boundary condition, as a function of the subsystem size ratio $r = \ell/L$. The continuous lines in the left (right) panel correspond to Eq. (67) for $\theta = 1/2$ ($\theta = 3/2$). The insets are a test for the imaginary part of the same quantity. The blue symbols are the numerical data obtained through the techniques explained in Appendix C.

Linear combinations of the above quantities are expected to reproduce the $n = 1$ charged moments of the linear combinations of operators $T \pm L_{-1}\partial\phi$, where $T$ is the stress-energy tensor and $L_{-1}\partial\phi$ is the Virasoro generator acting on the derivative field [68]. Indeed, we can compute the ratio between the excited and the ground state charged moments

$$
\begin{aligned}
\Delta Z_1(\theta) &\equiv \frac{Z_1^{\mathrm{Level\,2}}(\theta)}{Z_1^{\mathrm{gs}}(\theta)} - \frac{Z_1^{\mathrm{Level\,2}}(0)}{Z_1^{\mathrm{gs}}(0)} \\
&= \frac{1}{2}[F_1^{\mathrm{L}}(\theta; \{1,1\}, \{1,1\}; 0) + F_1^{\mathrm{L}}(\theta; \{2\}, \{2\}; 0)] + F_1^{\mathrm{L}}(\theta; \{1,1\}, \{2\}; 0) - 1 \\
&= -\frac{\theta^2\beta^2}{4\pi^2}\sin^2(\pi r)\left[-\frac{\theta^2\beta^2}{\pi^2}\sin^2(\pi r) + 4 + 4\cos^2(\pi r) + 2\mathrm{i}\frac{\beta\theta}{\pi}\sin(2\pi r)\right]. \quad (67)
\end{aligned}
$$

We have checked the result above against exact lattice computations in Figure 4, both for the real and imaginary part (see the inset). In the *XX* chain the CFT state corresponds to two states, one made up of two holes and one particle, and the other made up of one hole and two particles, and they both give the same contribution to the charged moments with $\beta = 1$. We refer to Appendix C for further details about the numerical techniques used to obtain the data in Figure 4.

As a final remark, in Figure 4 we observe small oscillations as a function of the parity of the subsystem size. This is reminiscent of the results found for the charged moments of the ground state of the XX spin chain in [10]. The authors found that the corrections to the leading order behavior of the charged moments (21) show an oscillatory behavior that depends both on the flux $\theta$ and on the Rényi index $n$ and they are amplified as both $n$ and $\theta$ increase. Thus, we expect that the oscillations we observe in Figure 4 are also due to the microscopic realization of the compact boson we are considering.

# 5 The $n = 2$ generalized charged moment

In this Section, we detail the computation of the normalized $n = 2$ generalized charged moment $F_2(\theta; \psi_1, \psi_2, \psi_3, \psi_4)$, paving the way to the computation of generalized second Rényi

entropy in Section 7. The input data here are four generic states $|\psi_i\rangle$ ($i = 1, ..., 4$) given by

$$|\psi_i\rangle = \mathcal{N}_i \prod_{j=1}^{N_{L_i}} a_{-k_{i,j}} \prod_{\bar{j}=1}^{N_{R_i}} \bar{a}_{-\tilde{k}_{i,\bar{j}}} |\alpha_i, \bar{\alpha}_i\rangle, \tag{68}$$

where the normalization factor $\mathcal{N}_i$ is given in Eq. (31). The vertex operator charges $\alpha_i, \bar{\alpha}_i$ need to satisfy the following conditions

$$\alpha_1 + \alpha_3 = \alpha_2 + \alpha_4, \quad \bar{\alpha}_1 + \bar{\alpha}_3 = \bar{\alpha}_2 + \bar{\alpha}_4, \tag{69}$$

to ensure a non-vanishing $n = 2$ generalized charged moment.

The normalized $n = 2$ generalized charged moment is defined as

$$
\begin{aligned}
F_2(\theta; \psi_1, \psi_2, \psi_3, \psi_4) &= \frac{\text{Tr}_A\left(\text{Tr}_{A^c}|\psi_3\rangle\langle\psi_4|\text{Tr}_{A^c}|\psi_1\rangle\langle\psi_2|e^{i\theta Q_A}\right)}{\text{Tr}_A\left(\text{Tr}_{A^c}^2|0\rangle\langle 0|e^{i\theta Q_A}\right)} \\
&= F_2^{\text{L}}(\theta; \psi_1, \psi_2, \psi_3, \psi_4) F_2^{\text{R}}(\theta; \psi_1, \psi_2, \psi_3, \psi_4). \tag{70}
\end{aligned}
$$

In the following, we will derive a sum formula for the normalized $n = 2$ generalized charged moment.

## 5.1 A sum formula for the $n = 2$ generalized charged moment

Due to the factorization between the left- and right-moving contributions in Eq. (70), here we will focus on the chiral/left-moving normalized $n = 2$ generalized charged moment, while the anti-chiral/right-moving contribution can be obtained via complex conjugation.

The chiral normalized $n = 2$ generalized charged moment is described by Eq. (38) with $n = 2$. For computational purposes, we perform the following conformal transformation from the cylindrical replica geometry to the planar replica geometry (see Figure 2):

$$w \longrightarrow \sqrt{\frac{\sin\frac{\pi(w-u)}{L}}{\sin\frac{\pi(w-v)}{L}}}. \tag{71}$$

We then have

$$F_2^{\text{L}}(\theta; \{k_{i,j}\}; \{\alpha_i\}) = \prod_{i=1}^{4} \mathcal{N}_i \left(-\frac{1}{2\pi}\right)^{N_{L_i}} e^{\frac{2\pi i}{L} v \sum_{j=1}^{N_{L_i}} \sigma_i k_{i,j}} \tag{72}$$

$$\times \prod_{j=1}^{N_{L_i}} \oint_{C_{\sigma_i}^{i,j}} dz_{i,j} \left(\frac{\partial z}{\partial w}\bigg|_{w_{i,0}} \left(\pm \frac{iL}{2\pi} e^{\pm\frac{2\pi i w_{i,0}}{L}}\right)\right)^{\frac{\alpha_i^2}{2}} \times \left(\frac{z_{i,j}^2 - (y_i^2)^*}{z_{i,j} + y_i} \frac{1}{z_{i,j} - y_i}\right)^{k_{i,j}}$$

$$\times \frac{\langle V_{\frac{\beta\theta}{2\pi}}(y_0) V_{-\frac{\beta\theta}{2\pi}}(y_{0'}) \prod_{i=1}^{4} V_{\pm\alpha_i}(y_i) \prod_{j=1}^{N_{L_1}} \partial\phi(z_{1,j}) \prod_{j=1}^{N_{L_2}} \partial\phi(z_{2,j}) \prod_{j=1}^{N_{L_3}} \partial\phi(z_{3,j}) \prod_{j=1}^{N_{L_4}} \partial\phi(z_{4,j})\rangle_{\mathbb{C}}}{\langle V_{\frac{\beta\theta}{2\pi}}(y_0) V_{-\frac{\beta\theta}{2\pi}}(y_{0'})\rangle_{\mathbb{C}}}.$$

The field insertions on the planar replica geometry are as illustrated in Figure 2. In particular, the four vertex operators are inserted at

$$y_1 = e^{\frac{i\pi r}{2}}, \qquad y_2 = e^{-\frac{i\pi r}{2}}, \qquad y_3 = -e^{\frac{i\pi r}{2}}, \qquad y_4 = -e^{-\frac{i\pi r}{2}}. \tag{73}$$

The integrals in Eq. (72) are performed along contours circling around $y_i$.

The correlation function ratio inside the integrand can be expanded as a sum according to different Wick contraction choices, in a fashion similar to Eq. (49) for the $n = 1$ generalized charged moment. Consequently, we can organize $F_2^{\mathrm{L}}(\theta; \{k_{i,j}\}; \{\alpha_i\})$ as an explicit sum:

$$F_2^{\mathrm{L}}(\theta; \{k_{i,j}\}; \{\alpha_i\}) = \prod_{i=1}^{4} \mathcal{N}_i \left(-\tfrac{1}{2\pi}\right)^M \xi(\{k_{i,j}\}) K(\alpha_i) \sum_{\substack{\{k_{m,n}\}, \\ \{k_{p,q}\}}} g_{V\partial\phi}(\theta, \{\alpha_i\}, \{k_{m,n}\}) g_{\partial\phi\partial\phi}(\{k_{p,q}\}), \quad (74)$$

where $M$ is the total number of (chiral) modes. In the following, we will describe in more details the various ingredients involved in this formula.

- The normalization factors $\mathcal{N}_i$ are given in Eq. (31). Similar to the case of $n = 1$ generalized charged moment, the phase factor $\xi(\{k_{i,j}\})$ is given by

$$\xi(\{k_{i,j}\}) = \exp\left[\frac{2\pi\mathrm{i}}{L} \nu \sum_{i=1}^{4} \sum_{j=1}^{N_{\mathrm{L}_i}} \sigma_i k_{i,j}\right]. \quad (75)$$

When restricting to the zero-momentum sector, this phase will be canceled by a similar phase coming from the anti-chiral/right-moving contributions.

- The prefactor $K(\alpha_i)$ corresponds to the contribution purely from the vertex operators. For notational convenience, we define $\alpha_1' = \alpha_1$, $\alpha_2' = -\alpha_2$, $\alpha_3' = \alpha_3$, and $\alpha_4' = -\alpha_4$. Then we have

$$K(\alpha_i) = \prod_{i=1}^{4} \langle V_{\frac{\beta\theta}{2\pi}}(y_0) V_{\alpha_i'}(y_i)\rangle \prod_{i=1}^{4} \langle V_{-\frac{\beta\theta}{2\pi}}(y_{0'}) V_{\alpha_i'}(y_i)\rangle \prod_{m>n=1}^{4} \langle V_{\alpha_m'}(y_m) V_{\alpha_n'}(y_n)\rangle$$

$$\times \left(\mathrm{e}^{\frac{2\pi\mathrm{i}}{L}\nu} \frac{\mathrm{i}\sin(\pi r)}{y_1}\right)^{\frac{\alpha_1^2}{2}} \left(\mathrm{e}^{-\frac{2\pi\mathrm{i}}{L}\nu} \frac{-\mathrm{i}\sin(\pi r)}{y_2}\right)^{\frac{\alpha_2^2}{2}} \left(\mathrm{e}^{\frac{2\pi\mathrm{i}}{L}\nu} \frac{\mathrm{i}\sin(\pi r)}{y_3}\right)^{\frac{\alpha_3^2}{2}} \left(\mathrm{e}^{-\frac{2\pi\mathrm{i}}{L}\nu} \frac{-\mathrm{i}\sin(\pi r)}{y_4}\right)^{\frac{\alpha_4^2}{2}} \quad (76)$$

$$= (-1)^{\alpha_2^2 - \alpha_3^2 + \frac{1}{2}(\alpha_1^2 - \alpha_3^2) + \alpha_1\alpha_2 - \alpha_1\alpha_2 - \alpha_1\alpha_3 + \alpha_2\alpha_3} \mathrm{e}^{\mathrm{i}\beta\theta r(\alpha_1 + \alpha_3)} \mathrm{e}^{\mathrm{i}\pi r(\alpha_2 - \alpha_1)(\alpha_2 - \alpha_3)}$$

$$\times \mathrm{e}^{\frac{\mathrm{i}\pi\nu}{L}(\alpha_1^2 + \alpha_3^2 - \alpha_2^2 - \alpha_4^2)} \left(\cos\frac{\pi r}{2}\right)^{(\alpha_2 - \alpha_3)^2} \left(\sin\frac{\pi r}{2}\right)^{(\alpha_1 - \alpha_2)^2},$$

where we have used $\sum_i \alpha_i' = \alpha_1 - \alpha_2 + \alpha_3 - \alpha_4 = 0$.

- The summation in Eq. (74) is organized based on different Wick contraction choices for the correlator ratio. Given the total collection $\{k_{i,j}\}$ where $i = 1, ..., 4$ labels the state $|\psi_i\rangle$ and $j = 1, ..., N_{\mathrm{L}_i}$ labels different creation modes $a_{-k_{i,j}}$ acting on the primary state in $|\psi_i\rangle$, we enumerate all possibilities to divide this set into two subsets $\{k_{m,n}\}$ and $\{k_{p,q}\}$. Elements in the first subset $\{k_{m,n}\}$ corresponds to $\partial\phi(z_{m,n})$ which are contracted with vertex operators. For the second subset $\{k_{p,q}\}$, the corresponding $\partial\phi(z_{p,q})$ are contracted between themselves.

- The function $g_{V\partial\phi}(\theta, \{\alpha_i\}, \{k_{m,n}\})$ contains contributions from Wick contractions of $\partial\phi$ with the vertex operators. Concretely it is given by

$$g_{V\partial\phi}(\theta, \{\alpha_i\}, \{k_{m,n}\}) = -\prod_{m,n} \sigma_m \left[\frac{2\pi\mathrm{i}}{(k_{m,n}-1)!} \partial_{z_{m,n}}^{k_{m,n}-1} \left(f(z_m, z_n)^{k_{m,n}} \frac{-\mathrm{i}\beta\theta}{2\pi z_{m,n}}\right)\Big|_{z_{m,n}=y_m}\right.$$

$$+ \sum_{\mu \neq m} \frac{2\pi\mathrm{i}}{(k_{m,n}-1)!} \partial_{z_{m,n}}^{k_{m,n}-1} \left(f(z_m, z_n)^{k_{m,n}} \frac{\mathrm{i}\alpha_\mu'}{y_\mu - z_{m,n}}\right)\Big|_{z_{m,n}=y_m} \quad (77)$$

$$\left.+ \frac{2\pi\mathrm{i}}{k_{m,n}!} \partial_{z_{m,n}}^{k_{m,n}} \left(-\mathrm{i}\alpha_m' f(z_m, z_n)^{k_{m,n}}\right)\Big|_{z_{m,n}=y_m}\right],$$

where

$$f(z_m, z_n) = \frac{z_{m,n}^2 - (y_m^2)^*}{z_{m,n} + y_m}.$$ (78)

In Eq. (77), $\sigma_m$ is again determined by the orientation of the corresponding integration contour.

- The function $g_{\partial\phi\partial\phi}(\{k_{p,q}\})$ encodes contributions from Wick contractions of $\partial\phi$ with themselves, which vanishes unless the set $\{k_{p,q}\}$ contains an even number of elements. Denoting the number of elements in the collection $\{k_{p,q}\}$ as $2l$, we first enumerate all possibilities of forming $l$ pairs $(k_{p_1,q_1}, k_{r_1,s_1}), ..., (k_{p_l,q_l}, k_{r_l,s_l})$. The contribution from each pairing possibility is the product over contributions from each individual pair $(k_{p_i,q_i}, k_{r_i,s_i})$. Namely

$$g_{\partial\phi\partial\phi}(\{k_{p,q}\}) = \sum_{\substack{\{(k_{p_1,q_1}, k_{r_1,s_1}) \\ ...(k_{p_l,q_l}, k_{r_l,s_l})\}}} \prod_{(k_{p_i,q_i}, k_{r_i,s_i})} d(p_i, r_i, k_{p_i,q_i}, k_{r_i,s_i}),$$ (79)

where $d(p_i, r_i, k_{p_i,q_i}, k_{r_i,s_i})$ contains contribution from the following two-point function $\langle \partial\phi(z_{p_i,q_i}) \partial\phi(z_{r_i,s_i}) \rangle_{\mathbb{C}}$. The function $d(p_i, r_i, k_{p_i,q_i}, k_{r_i,s_i})$ can be computed in the same way as for the case of $n = 1$ generalized charged moment, see equations (A.10-A.12).

Unlike the case of $n = 1$ generalized charged moment, Eq. (77) and Eq. (79) generically do not admit further simplifications, except for when the subsystem size is exactly half of the total system size. We spell out the simplification for this special case in Appendix A.2.

### 5.1.1 The normalized $n = 2$ generalized charged moment as an explicit sum

We can write the result (72) in a form similar to Eq. (58), by introducing

$$R_{k_1,...,k_M} = \prod_{i=1}^{4} \mathcal{N}_i \left( -\frac{1}{2\pi} \right)^M g_{\partial\phi\partial\phi}(\{k_{p,q}\}).$$ (80)

The main difference is that the structure is more complex because a simplification like the one in Eq. (53) does not occur. This means that all the contractions between derivative and vertex operators contribute to the final result, and we get

$$F_2^{\mathrm{L}}(\theta; \{k_{i,j}\}; \{\alpha_i\}) = K(\alpha_i)\xi(\{k_{i,j}\})\Big[ R_{k_1,...,k_M} + \sum_{i=1}^{M} R_{k_1,...,\not{k}_i...k_M} N(k_i) L_{k_i}(\theta)$$ (81)

$$+ \sum_{i_1 < i_2}^{M} R_{k_1,...,\not{k}_{i_1}...,\not{k}_{i_2},...k_M} N(k_{i_1}, k_{i_2}) L_{k_{i_1}}(\theta) L_{k_{i_2}}(\theta) + ...$$

$$+ \prod_{i=1}^{M} L_{k_i}(\theta) + \sum_{i=1}^{M} R_{k_1,...,\not{k}_i...k_N} N(k_i) \tilde{L}_{k_i}(\bar{\alpha})$$

$$+ \sum_{i_1 < i_2}^{M} R_{k_1,...,\not{k}_{i_1}...,\not{k}_{i_2},...k_N} N(k_{i_1}, k_{i_2}) \tilde{L}_{k_{i_1}}(\bar{\alpha}) \tilde{L}_{k_{i_2}}(\bar{\alpha}) + ...$$

$$+ \prod_{i=1}^{M} L_{k_i}(\bar{\alpha}) + \sum_{i_1 < i_2}^{M} R_{k_1,...,\not{k}_{i_1}...,\not{k}_{i_2},...k_M} N(k_{i_1}, k_{i_2})$$

$$\times (L_{k_{i_1}}(\theta) \tilde{L}_{k_{i_2}}(\bar{\alpha}) + L_{k_{i_2}}(\theta) \tilde{L}_{k_{i_1}}(\bar{\alpha}))$$

$$+ \cdots + \sum_{K_1, K_2} \prod_{i \in K_1} \prod_{i \in K_2} L_{k_{j \in K_1}}(\theta) \tilde{L}_{k_{j \in K_2}}(\bar{\alpha}) \Big].$$

Here $K_1$ and $K_2$ are two arbitrary sets of the momenta $\{k_j\}$ such that $K_1 \cup K_2$ contains $k_1, \ldots, k_M$, $\bar{\alpha} = \{\alpha_1, \alpha_2, \alpha_3, \alpha_4\}$, and we have introduced the functions

$$L_{k_j}(\theta) = \frac{\beta\theta}{2\pi} \sigma_j \frac{1}{\Gamma(k_j)} \partial_{z_j}^{k_j-1} \frac{f^{k_j}(z_j, k_j)}{z_j}, \tag{82}$$

which describes the contraction between $\partial\phi$ and $V_{\frac{\beta\theta}{2\pi}}$, and

$$\tilde{L}_{k_j}(\bar{\alpha}) = \sum_{i=1}^{4} \alpha_i \sigma_j J_{ij},$$

$$J_{ij} = \begin{cases} \frac{1}{\Gamma(k_j+1)} \partial_{z_j}^{k_j} f^{k_j}(z_j, k_j), & i = j, \\ \frac{1}{\Gamma(k_j)} \partial_{z_j}^{k_j-1} \frac{f^{k_j}(z_j, k_j)}{z_j - y_i}, & i \neq j, \end{cases} \tag{83}$$

which encodes the contraction between $\partial\phi$ and $V_{\pm\alpha_i}$.

## 5.2 Examples and benchmarks of the results

We now use the formula (81) to compute the $n = 2$ generalized charged moments and benchmark them with known results. As a first example, starting from the chiral states

$$|\psi_1\rangle = a_{-1}|0\rangle, \quad |\psi_2\rangle = a_{-1}|0\rangle, \quad |\psi_3\rangle = |\alpha\rangle, \quad |\psi_4\rangle = |\alpha\rangle, \tag{84}$$

we can compute the charged moments for the second relative entropy between $\partial\phi$ and $V_\alpha$ previously computed in [20]. Indeed, using our procedure, we find

$$\frac{F_2^{\mathrm{L}}(\theta; \{1\}, \{1\}, \{0\}, \{0\}; \{0, 0, \alpha, \alpha\})}{F_2^{\mathrm{L}}(\theta; \{0\}, \{0\}, \{0\}, \{0\}; \{0, 0, \alpha, \alpha\})} = e^{i\frac{r}{2}\theta\beta\alpha} \left[ \frac{\beta\theta^2}{4\pi^2} \sin(\pi r)^2 + i\frac{\beta\theta\alpha}{2\pi} \sin\left(\frac{\pi r}{2}\right)^2 \sin(\pi r) \right.$$

$$\left. + \frac{1 + \cos(\pi r)}{2} + \alpha^2 \sin\left(\frac{\pi r}{2}\right)^4 \right], \tag{85}$$

which reproduces the result found in [20]. As a further check, by setting $\alpha = 0$, we retrieve the charged moment for the second relative entropy between $\partial\phi$ and the ground state [20]

$$F_2^{\mathrm{L}}(\theta; \{1\}, \{1\}, \{0\}, \{0\}; \{0, 0, 0, 0\}) = 1 - 1\frac{\beta^2\theta^2 \sin^2\left(\frac{\pi r}{2}\right)}{\pi^2}. \tag{86}$$

Finally, we can consider

$$|\psi_1\rangle = a_{-1}|0\rangle, \quad |\psi_2\rangle = a_{-1}|0\rangle, \quad |\psi_3\rangle = a_{-1}|0\rangle, \quad |\psi_4\rangle = a_{-1}|0\rangle, \tag{87}$$

which gives the $n = 2$ charged moment for the excited state $\partial\phi$:

$$F_2^{\mathrm{L}}(\theta; \{1\}, \{1\}, \{1\}, \{1\}; \{0, 0, 0, 0\}) = \beta^4\theta^4 \frac{\sin^4(\pi r)}{16\pi^4} + \beta^2\theta^2 \frac{1}{16\pi^2} \sin^2(\pi r)[\cos(2\pi r) - 9]$$

$$+ \frac{1}{64} \left[\cos(2\pi r) + 7\right]^2. \tag{88}$$

This result also matches with the one found in [20].

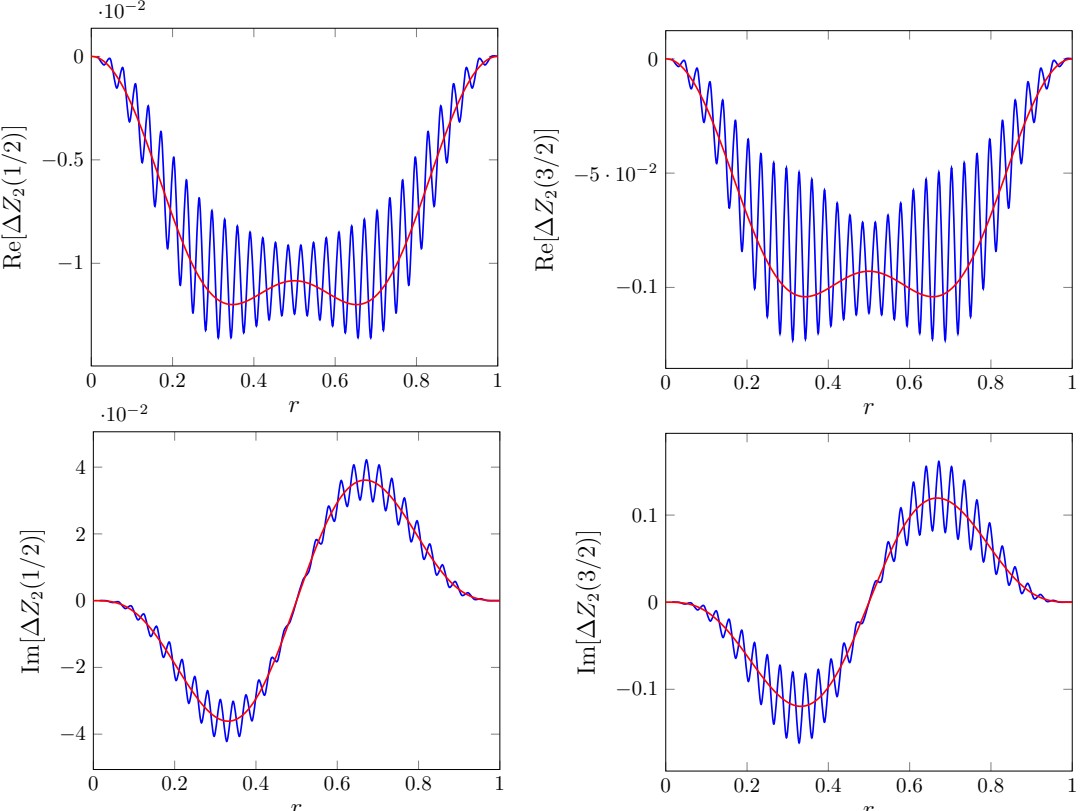

Figure 5: The top (bottom) panels are a test of the real (imaginary) part of the CFT result for the $n = 2$ generalized charged moments in Eq. (89). The blue lines correspond to numerical data for the XX spin chain of size $L = 64$ with periodic boundary condition and the red lines correspond to predictions from Eq. (89). The data are plotted as a function of the subsystem ration $r$ for two different values of $\theta$.

### 5.2.1 The $n = 2$ generalized charged moments involving level-2 chiral states

We can now perform an analysis, similar to that of Section 4.2.2, to compute the $n = 2$ generalized charged moments at chiral conformal level-2. Starting from the states (65) where the vertex operator charge is set to 0, we collect a list of all the level-2 chiral charged moments in Appendix B.

As we did in Section 4.2.2, we group these results to obtain the normalized charged moment for the resolved second Rényi entropy for states corresponding to $T \pm L_{-1}\partial\phi$, and we get

$$
\begin{aligned}
\Delta Z_2(\theta) &\equiv \frac{Z_2^{\text{Level 2}}(\theta)}{Z_2^{\text{gs}}(\theta)} - \frac{Z_2^{\text{Level 2}}(0)}{Z_2^{\text{gs}}(0)} \\
&= \Big[ F_2^{\text{L}}(\theta; \{1,1\}, \{1,1\}, \{1,1\}, \{2\}) + F_2^{\text{L}}(\theta; \{1,1\}, \{2\}, \{2\}, \{2\}) \\
&\quad + \frac{1}{4} \big( 2F_2^{\text{L}}(\theta; \{1,1\}, \{2\}, \{2\}, \{1,1\}) + 2F_2^{\text{L}}(\theta; \{1,1\}, \{1,1\}, \{2\}, \{2\}) \\
&\quad\quad + 2F_2^{\text{L}}(\theta; \{1,1\}, \{2\}, \{1,1\}, \{2\}) + F_2^{\text{L}}(\theta; \{1,1\}, \{1,1\}, \{1,1\}, \{1,1\}) \\
&\quad\quad + F_2^{\text{L}}(\theta; \{2\}, \{2\}, \{2\}, \{2\}) \big) \Big] \Big|_{\theta=0}^{\theta}
\end{aligned}
\tag{89}
$$

$$= \frac{\beta^8\theta^8}{4096\pi^8}\sin^8(\pi r) - \frac{\beta^6\theta^6}{2048\pi^6}(41+23\cos(2\pi r))\sin^6(\pi r)$$

$$+ \frac{\beta^4\theta^4}{16384\pi^4}(3129+1460\cos(2\pi r)+19\cos(4\pi r))\sin^4(\pi r)$$

$$+ \frac{\beta^2\theta^2}{32768\pi^2}(-24826-9417\cos(2\pi r)+1386\cos(4\pi r)+89\cos(6\pi r))\sin^2(\pi r)$$

$$- \frac{i\theta\sin^3(\pi r)\cos(\pi r)}{2048\pi}\left[\frac{3\theta^6}{\pi^6}-\frac{89\theta^4}{\pi^4}+\frac{681\theta^2}{\pi^2}+\left(1-\frac{\theta}{\pi^2}\right)^2\left(\frac{\theta}{\pi^2}+1\right)^2\right.$$

$$\left.\times\left(\left(68-\frac{4\theta^2}{\pi^2}\right)\cos(2\pi r)+\left(\frac{\theta^2}{\pi^2}+15\right)\cos(4\pi r)\right)+1453\right].$$

This expression has been checked against numerical computations in Figure 5 for the chiral component of the bosonic field when $\beta = 1$, $\theta = 1/2$, and $\theta = 3/2$. We notice that the oscillatory behavior of Figure 5 is more evident than in Figure 4, thus confirming our idea that the oscillations are due to lattice details, and they are amplified as $\theta$ and $n$ increase, in agreement with the findings of [10].

## 6 The generalized subsystem charge distribution

As we have explained in the Introduction, a basic building block in computing the time evolution of full counting statistics is the generalized subsystem charge distribution, defined as

$$P(q;\psi_1,\psi_2) = \int_{-\pi}^{\pi}\frac{d\theta}{2\pi}e^{-i\theta q}\mathrm{Tr}_A\left(\mathrm{Tr}_{A^c}|\psi_1\rangle\langle\psi_2|e^{i\theta Q_A}\right)$$
$$= \int_{-\pi}^{\pi}\frac{d\theta}{2\pi}e^{-i\theta q}F_1(\theta;\psi_1,\psi_2)\mathrm{Tr}_A\left(\mathrm{Tr}_{A^c}|0\rangle\langle 0|e^{i\theta Q_A}\right), \tag{90}$$

where $F_1(\theta;\psi_1,\psi_2)$ is the normalized $n=1$ generalized charged moment.

For out-of-equilibrium settings such as global quenches, suitable post-quench observables are defined relative to their pre-quench values. Therefore, a useful quantity to consider is the ground state normalized generalized subsystem charge distribution

$$P^{\mathrm{rel}}(q;\psi_1,\psi_2) := \frac{P(q;\psi_1,\psi_2)}{P(q;|0\rangle,|0\rangle)} = \frac{\int_{-\pi}^{\pi}\frac{d\theta}{2\pi}e^{-i\theta q}F_1(\theta;\psi_1,\psi_2)\mathrm{Tr}_A\left(\mathrm{Tr}_{A^c}|0\rangle\langle 0|e^{i\theta Q_A}\right)}{\int_{-\pi}^{\pi}\frac{d\theta}{2\pi}e^{-i\theta q}\mathrm{Tr}_A\left(\mathrm{Tr}_{A^c}|0\rangle\langle 0|e^{i\theta Q_A}\right)}. \tag{91}$$

We recall that to leading order in the chord length $\ell$, $F_1(\theta;\psi_1,\psi_2)$ is a finite polynomial in $\theta$. To compute $P^{\mathrm{rel}}(q;\psi_1,\psi_2)$, we explicitly perform the numerator Fourier transformation at each order in $\theta$ in a similar way as in Eq. (22). The dependence on the entanglement cut boundary conditions cancels out between the numerator and the denominator.

As discussed in Section 4, $F_1(\theta;\psi_1,\psi_2)$ is a polynomial in $\theta$ in which all the powers of $\theta$ are either exclusively even or exclusively odd. We focus on the case where $F_1(\theta;\psi_1,\psi_2)$ only contains even powers of $\theta$, while a complete analysis involving the odd power case will be given elsewhere. In practice, the coefficients of the polynomial decrease as the power of $\theta$ increases, such that the result can be truncated at a certain order in $\theta$. Here we choose the truncation at $o(\theta^4)$, namely we have

$$F_1(\theta;\psi_1,\psi_2) = 1 + \theta^2\beta^2 h_2 + \theta^4\beta^4 h_4 + O(\theta^6), \tag{92}$$

where $h_2$ and $h_4$ are trigonometric functions in the subsystem size ratio $r$. We can then compute the normalized generalized subsystem charge distribution as an expansion in $\frac{1}{\log\ell'}$, where

$\ell' := \ell/\epsilon$. Concretely, we find

$$
\begin{aligned}
P^{\text{rel}}(q; \psi_1, \psi_2) &= 1 + h_2 \frac{\pi^2(\beta^2 \log\ell' - \pi^2 q^2)}{(\beta \log\ell')^2} \\
&\quad + h_4 \frac{\pi^4\left(3(\beta^2\log\ell')^2 - 6\pi^2 q^2 \beta^2 \log\ell' + \pi^4 q^4\right)}{(\beta\log\ell')^4} + O\left(\frac{1}{(\log\ell')^3}\right) \\
&= 1 + h_2 \frac{\pi^2}{\log\ell'} + \left(3h_4 - \frac{h_2}{\beta^2}q^2\right)\frac{\pi^4}{(\log\ell')^2} + O\left(\frac{1}{(\log\ell')^3}\right).
\end{aligned} \tag{93}
$$

This then yields the generalized subsystem charge distribution $P(q; \psi_1, \psi_2)$ as a modified distribution (compared with the ground state Gaussian distribution), where the correction is given as an expansion in $1/\log\ell'$.

We can also directly approximate $P(q; \psi_1, \psi_2)$ in a more compact form which indicates more clearly the correction to the variance of the subsystem charge distribution. Using the Gaussian approximation

$$
F_1(\theta; \psi_1, \psi_2) \sim e^{h_2 \beta^2 \theta^2}, \tag{94}
$$

we have

$$
P(q; \psi_1, \psi_2) \sim \sqrt{\frac{\pi}{2\beta^2 b_1}} e^{-\frac{\pi^2 q^2}{2\beta^2 b_1}}, \tag{95}
$$

where the shifted variance is given by $\beta^2 b_1$ with

$$
b_1 := \log\ell' - 2\pi^2 h_2. \tag{96}
$$

We recall that, in the ground state, the variance is simply given by $\beta^2 \log\ell'$. However, here we find a universal shift arising purely from the fact that we are considering the generalized subsystem charge distribution involving excited states. Such a shift in the variance of the charge distribution has also been observed in other contexts. For instance, in [10], the charged moments were computed for the one-dimensional tight-binding model and the shift in variance is due to lattice contributions, which can be exactly calculated in a free fermionic system. Another setup where the shift in variance is universal and arises solely from field theory contributions was studied in [45]. In this case, by analyzing symmetry resolution in a massive field theory, the corrections to the variance remain universal and are due to the presence of a mass.

# 7 Symmetry-resolved generalized second Rényi entropy

In this Section, we outline the details involved in the computation of the symmetry-resolved generalized second Rényi entropy in the charge-$q$ sector, together with its application in the computation of the excited state symmetry-resolved second Rényi entropy.

The symmetry-resolved generalized second Rényi entropy, as defined in Eq. (3), can be expressed as

$$
S_2(q; \psi_1, \psi_2, \psi_3, \psi_4) = -\log \frac{\int_{-\pi}^{\pi} \frac{d\theta}{2\pi} e^{-i\theta q} F_2(\theta; \psi_1, \psi_2, \psi_3, \psi_4) \text{Tr}_A\left(\text{Tr}_{A^c}^2 |0\rangle\langle 0| e^{i\theta Q_A}\right)}{\int_{-\pi}^{\pi} \frac{d\theta}{2\pi} e^{-i\theta q} \text{Tr}_A\left(\text{Tr}_{A^c}^2 |0\rangle\langle 0| e^{i\theta Q_A}\right)}, \tag{97}
$$

where $F_2(\theta; \psi_1, \psi_2, \psi_3, \psi_4)$ is the normalized $n = 2$ generalized charged moment discussed in Section 5. To evaluate $S_2(q; \psi_1, \psi_2, \psi_3, \psi_4)$, we adopt the same strategy as that used for the generalized subsystem charge distribution. To leading order in the chord length $\ell$,

$F_2(\theta; \psi_1, \psi_2, \psi_3, \psi_4)$ is a finite polynomial in $\theta$, enabling us to perform the Fourier transformation at each order in $\theta$ similar to Eq. (22). Here we again focus on the case where the normalized generalized charged moments only contain even powers of $\theta$. To illustrate the computation, we truncate the polynomials at $o(\theta^4)$, as the coefficients of higher-order terms in $\theta$ are significantly smaller.

Taking the following expansion

$$F_2(\theta; \psi_1, \psi_2, \psi_3, \psi_4) = f_0 + \theta^2 \beta^2 f_2 + \theta^4 \beta^4 f_4 + O(\theta^6), \tag{98}$$

we can write $S_2(q; \psi_1, \psi_2, \psi_3, \psi_4)$ as an expansion in $\frac{1}{\log \ell'}$ as follows:

$$S_2(q; \psi_1, \psi_2, \psi_3, \psi_4) = -\log f_0 - \frac{2f_2}{f_0} \frac{\pi^2}{\log \ell'} + \left( \frac{2f_2^2}{f_0^2} - \frac{12f_4}{f_0} + \frac{4f_2}{f_0 \beta^2} q^2 \right) \frac{\pi^4}{(\log \ell')^2} + O\left( \frac{1}{(\log \ell')^3} \right). \tag{99}$$

As previously discussed, the symmetry-resolved generalized Rényi entropies are the fundamental building blocks for calculating the symmetry resolution of entanglement in excited states. Here we consider the symmetry-resolved second Rényi entropy for an arbitrary excited state $|\psi\rangle$, defined as

$$\mathcal{S}_2(q; \psi) = -\log \frac{\text{Tr}_A \left[ \text{Tr}_{A^c} (|\psi\rangle\langle\psi|) \text{Tr}_{A^c} (|\psi\rangle\langle\psi|) \Pi_q \right]}{\text{Tr}_A^2 \left[ \text{Tr}_{A^c} (|\psi\rangle\langle\psi|) \Pi_q \right]}. \tag{100}$$

In particular, the difference between the excited state and the ground state symmetry resolution is given by

$$\Delta \mathcal{S}_2(q; \psi) := \mathcal{S}_2(q; \psi) - \mathcal{S}_2(q; |0\rangle) = S_2(q; \psi, \psi, \psi, \psi) + 2\log P^{\text{rel}}(q; \psi, \psi),$$

where $P^{\text{rel}}(q; \psi, \psi)$ is the normalized excited state subsystem charge distribution obtained by setting $\psi_1 = \psi_2 = \psi$ in Eq. (91).

We now evaluate $\Delta \mathcal{S}_2(q; \psi)$ in a similar fashion as before. Concretely, taking

$$F_2(\theta; \psi, \psi, \psi, \psi) = f_0 + \theta^2 \beta^2 f_2 + \theta^4 \beta^4 f_4 + O(\theta^6),$$
$$F_1(\theta; \psi, \psi) = 1 + \theta^2 \beta^2 h_2 + \theta^4 \beta^4 h_4 + O(\theta^6), \tag{101}$$

the difference between the excited state and the ground state symmetry-resolved second Rényi entropies can be written as

$$\Delta \mathcal{S}_2(q; \psi) = -\log f_0 + 2\left( h_2 - \frac{f_2}{f_0} \right) \frac{\pi^2}{\log \ell'} \tag{102}$$
$$+ 2\left[ \frac{f_2^2}{f_0^2} - \frac{6f_4}{f_0} - \frac{h_2^2}{2} + 3h_4 + \left( \frac{2f_2}{f_0 \beta^2} - \frac{h_2}{\beta^2} \right) q^2 \right] \frac{\pi^4}{(\log \ell')^2} + O\left( \frac{1}{(\log \ell')^3} \right).$$

In particular, the dependence on the subsystem charge $q$ only appears at $O((\log \ell')^{-2})$, and this resembles once again the terms breaking the equipartition of the entanglement found in [10], in a lattice context, or in [18], by studying the symmetry resolution of excited states in CFT.

We can also directly perform an approximate evaluation of the excited state symmetry-resolved second Rényi entropy $\mathcal{S}_2(q; \psi)$, by approximating the charged moments as

$$F_2(\theta; \psi, \psi, \psi, \psi) \sim f_0 e^{\frac{f_2}{f_0} \beta^2 \theta^2},$$
$$F_1(\theta; \psi, \psi) \sim e^{h_2 \beta^2 \theta^2}. \tag{103}$$

This rewriting allows us to express the excited state symmetry-resolved second Rényi entropy in a compact form as

$$
\begin{aligned}
\mathbb{S}_2(q;\psi) = {} & \frac{1}{4}\log\ell' - \frac{1}{2}\log\log(\ell'\delta) - \log\frac{2\beta f_0}{\sqrt{\pi}} + 2\log g_a \\
& - 2\frac{\pi^4(2f_2/f_0 - h_2)^2}{(\log\ell')^2} + \frac{2\pi^4 q^2(h_2 - 2f_2/f_0)}{\beta^2(\log(\ell'\kappa))^2} + O\left(\frac{1}{(\log\ell')^2}\right),
\end{aligned} \tag{104}
$$

where $g_a$ is the entanglement boundary $g$-function and

$$
\delta = e^{-4\pi^2(h_2 - f_2/f_0)}, \qquad \kappa = e^{-\pi^2(h_2 + 2f_2/f_0)}.
$$

Beyond the breaking of the equipartition at $O((\log\ell')^{-2})$, this rewriting shows the presence of a double logarithmic correction, which is another salient feature of the ground state symmetry resolution [8, 10].

## 8 Conclusions

In this manuscript, we have introduced the notion of symmetry-resolved generalized entropies, which form important computational building blocks in the study of symmetry-resolved entanglement for generic excited states as well as the dynamical evolution of symmetry-resolved entanglement in symmetry-preserving out-of-equilibrium settings. A crucial ingredient in the computation of symmetry-resolved generalized entropies is the generalized charged moments, which also have their own physical meaning. In particular, the $n = 1$ generalized charged moment is an essential ingredient for the study of the probability distribution of the symmetry charge contained within a subsystem, giving access to the corresponding full counting statistics.

We have illustrated our computational framework in the example of (1+1)-d free massless compact boson theory, motivated by the experimental accessibility of Luttinger liquids. Using the replica-trick computation, we have provided explicit analytical formulas for the $n = 1$ and the $n = 2$ generalized charged moments and, when possible, benchmarked our results through lattice computations in the XX chain. Furthermore, we have outlined the computation of generalized subsystem charge distribution and symmetry-resolved generalized second Rényi entropy, where we have observed universal subleading terms breaking the entanglement equipartition.

As already mentioned, the most direct and appealing application of our results would be the study of the full counting statistics or the symmetry-resolved entanglement after a global quantum quench, especially in those setups where a quasi-particle picture or other tools are not available [84]. Given the connections of the quantities studied here with the pseudo-entropies introduced in [79, 80], an interesting further possibility would be to compare our results with the holographic findings, also in the absence of global conserved charges. Finally, explicit results could be also derived for other CFTs with different global symmetries, such as the Ising model with a $\mathbb{Z}_2$ symmetry, or further extended away of criticality by applying conformal perturbation theory.

## Acknowledgments

We thank Luca Capizzi and Giuseppe Di Giulio for helpful discussions related to this project. FY thanks the Simons Center for Geometry and Physics at Stony Brook University for great hospitality at various stages of this work.

**Funding information** The work of FY was supported by the U.S. Department of Energy, Office of Basic Energy Sciences, under Contract No. DE-SC0012704. SM thanks the support from the Walter Burke Institute for Theoretical Physics and the Institute for Quantum Information and Matter at Caltech. PC acknowledges support from ERC under Consolidator Grant number 771536 (NEMO) and from European Union - NextGenerationEU, in the framework of the PRIN Project HIGHEST no. 2022SJCKAH_002.

# A  Computational ingredients for generalized charged moments

In this Appendix, we spell out some detailed ingredients involved in the evaluation of normalized generalized charged moments.

## A.1  The chiral normalized $n = 1$ generalized charged moment

The chiral normalized $n = 1$ generalized charged moment, $F_1^{\mathrm{I}}(\theta; \psi_1, \psi_2)$, is given by Eq. (50). Here the main computational ingredients are the functions $g_{V\partial\phi}(\theta, \alpha, \{k_{m,n}\})$ and $g_{\partial\phi\partial\phi}(\{k_{p,q}\})$.

The function $g_{V\partial\phi}(\theta, \alpha, \{k_{m,n}\})$ encodes contributions from Wick contractions of $\partial\phi(z_{m,n})$ with the vertex operators. Concretely, each contour integral of $\partial\phi(z_{m,n})$ contributes the following:

$$\oint_{C^{m,n}} dz_{m,n} \left( \frac{z_{m,n} - y_m^*}{z_{m,n} - y_m} \right)^{k_{m,n}} \sum_{k=0,1,2} \langle V_{\alpha_k}(y_k) \partial\phi(z_{m,n}) \rangle_{\mathbb{C}}, \tag{A.1}$$

where $C^{m,n}$ are contours around $y_m (m = 1, 2)$ (see Figure 3 for more details) and $\alpha_0 = \beta\theta/(2\pi)$, $\alpha_1 = -\alpha_2 = \alpha$. We then have

$$g_{V\partial\phi}(\theta, \alpha, \{k_{m,n}\}) = - \prod_{(m,n)} \sigma_m \times \left( g_{V\partial\phi}^{(1)}(m, k_{m,n}, \theta) + g_{V\partial\phi}^{(2)}(m, k_{m,n}, \alpha) \right). \tag{A.2}$$

Here $\sigma_m$ is as defined in Eq. (38) with $\sigma_{m=1} = 1$ and $\sigma_{m=2} = -1$. The functions $g_{V\partial\phi}^{(1),(2)}$ are given by

$$g_{V\partial\phi}^{(1)}(m, k_{m,n}, \theta) := \frac{2\pi i}{(k_{m,n} - 1)!} \partial_{z_{m,n}}^{k_{m,n}-1} \left( f(m,n)^{k_{m,n}} \times \left( -\frac{i\beta\theta}{2\pi z_{m,n}} \right) \right) \Bigg|_{z_{m,n}=y_m}, \tag{A.3}$$

$$\begin{aligned} g_{V\partial\phi}^{(2)}(m, k_{m,n}, \alpha) := &\sum_{\mu \neq m} \frac{2\pi i}{(k_{m,n} - 1)!} \partial_{z_{m,n}}^{k_{m,n}-1} \left( f(m,n)^{k_{m,n}} \times \left( \frac{i\alpha_\mu}{y_\mu - z_{m,n}} \right) \right) \Bigg|_{z_{m,n}=y_m} \\ &+ \frac{2\pi i}{k_{m,n}!} \partial_{z_{m,n}}^{k_{m,n}} \left( -i\alpha_m f(m,n)^{k_{m,n}} \right) \Bigg|_{z_{m,n}=y_m}, \end{aligned} \tag{A.4}$$

with

$$f(m,n) = z_{m,n} - y_m^*. \tag{A.5}$$

Explicit calculation then yields

$$\begin{aligned} g_{V\partial\phi}^{(1)}(m = 1, k_{1,n}, \theta) &= \beta\theta \left( 1 - e^{-2\pi i r k_{1,n}} \right), \\ g_{V\partial\phi}^{(1)}(m = 2, k_{2,n}, \theta) &= \beta\theta \left( 1 - e^{2\pi i r k_{2,n}} \right), \\ g_{V\partial\phi}^{(2)}(m, k_{m,n}, \alpha) &= 2\pi(\alpha - \alpha) = 0. \end{aligned} \tag{A.6}$$

Putting everything together, we have

$$g_{V\partial\phi}(\theta, \alpha, \{k_{m,n}\}) = -\prod_{(m,n)} \sigma_m \beta \theta \left(1 - e^{\text{sign}(m)2\pi i r k_{m,n}}\right). \tag{A.7}$$

We now turn to $g_{\partial\phi\partial\phi}(\{k_{p,q}\})$ which corresponds to the contribution where $\partial\phi$ contract between themselves. In particular, $g_{\partial\phi\partial\phi}(\{k_{p,q}\})$ is zero unless the set $\{k_{p,q}\}$ contains an even number of elements. Given the collection $\{k_{p,q}\}$ with $2l$ elements, we first enumerate all possibilities of forming $l$ pairs $(k_{p_1,q_1}, k_{r_1,s_1}), ..., (k_{p_l,q_l}, k_{r_l,s_l})$. The contribution from each pair-up possibility is then given by the product over contributions from each invidual pair $(k_{p_i,q_i}, k_{r_i,s_i})$. Namely

$$g_{\partial\phi\partial\phi}(\{k_{p,q}\}) = \sum_{\substack{\{(k_{p_1,q_1}, k_{r_1,s_1}) \\ ...(k_{p_l,q_l}, k_{r_l,s_l})\}}} \prod_{(k_{p_i,q_i}, k_{r_i,s_i})} d(p_i, r_i, k_{p_i,q_i}, k_{r_i,s_i}), \tag{A.8}$$

where $d(p_i, r_i, k_{p_i,q_i}, k_{r_i,s_i})$ captures the contribution from the following two-point function $\langle \partial\phi(z_{p_i,q_i})\partial\phi(z_{r_i,s_i})\rangle_{\mathbb{C}}$. Concretely it is given by

$$d(p, r, k_{p,q}, k_{r,s}) = \oint_{C^{p,q}} dz_{p,q} \oint_{C^{r,s}} dz_{r,s} \left(\frac{z_{p,q} - y_p^*}{z_{p,q} - y_p}\right)^{k_{p,q}} \left(\frac{z_{r,s} - y_r^*}{z_{r,s} - y_r}\right)^{k_{r,s}} \frac{-1}{(z_{p,q} - z_{r,s})^2}. \tag{A.9}$$

We define an intermediate function $h(z_{p,q}, r, k_{r,s})$ as

$$h(z_{p,q}, r, k_{r,s}) = -\sigma_r \frac{2\pi i}{(k_{r,s} - 1)!} \partial_{z_{r,s}}^{k_{r,s}-1} \left(-\frac{1}{(z_{p,q} - z_{r,s})^2} f(r,s)^{k_{r,s}}\right)\Bigg|_{z_{r,s}=y_r}. \tag{A.10}$$

Then when $p \neq r$, namely when the two integration contours in Eq. (A.9) circle around two different poles, we have

$$d(p, r, k_{p,q}, k_{r,s}) = -\sigma_p \frac{2\pi i}{(k_{p,q} - 1)!} \partial_{z_{p,q}}^{k_{p,q}-1} \left(f(p,q)^{k_{p,q}} h(z_{p,q}, r, k_{r,s})\right)\Bigg|_{z_{p,q}=y_p}. \tag{A.11}$$

When $p = r$, we then have

$$d(p, r = p, k_{p,q}, k_{r,s}) = -\sigma_p \frac{2\pi i}{(k_{p,q} + k_{r,s})!} \partial_{z_{p,q}}^{k_{p,q}+k_{r,s}} \left((z_{p,q} - y_p)^{k_{r,s}+1} f(p,q)^{k_{p,q}} h(z_{p,q}, r, k_{r,s})\right)\Bigg|_{z_{p,q}=y_p}. \tag{A.12}$$

Explicit calculation with $y_1 = e^{i\pi r}$ and $y_2 = e^{-i\pi r}$ then gives

$$d(p, r, k_{p,q}, k_{r,s}) = -4\pi^2 \sigma_p \sigma_r e(p, r, k_{p,q}, k_{r,s}), \tag{A.13}$$

where

$$\begin{aligned} e(1, 2, k_1, k_2) &= k, \quad \text{if } k_1 = k_2 = k, \\ &= 0, \quad \text{if } k_1 \neq k_2, \end{aligned} \tag{A.14}$$

and $e(1, 1, k_1, k_2) = e(2, 2, k_1, k_2) = 0$. In other words, contribution from the two-point function $\langle \partial\phi(z_{p,q})\partial\phi(z_{r,s})\rangle_{\mathbb{C}}$ vanishes, unless the integration contours for $z_{p,q}$ and $z_{r,s}$ circle around different poles, and their integrands has the same pole order.

## A.2 Simplification of the chiral normalized $n = 2$ generalized charged moment at $r = \frac{1}{2}$

The normalized $n = 2$ generalized charged moment admits a simplified expression when the subsystem size is half of the total system size, namely when $r = \frac{1}{2}$.

Without loss of generality, we consider the chiral normalized $n = 2$ generalized charged moment $F_2^L(\theta; \{k_{i,j}\}; \{\alpha_i\})$, given in Eq. (74). The main ingredients in computing this quantity are the functions $g_{V\partial\phi}(\theta, \{\alpha_i\}, \{k_{m,n}\})$ encoding contributions from Wick contractions of $\partial\phi$ with the vertex operators, and $g_{\partial\phi\partial\phi}(\{k_{p,q}\})$ corresponding to contributions from Wick contractions of $\partial\phi$ with themselves. Our goal here is to find simplified expressions for these functions at $r = \frac{1}{2}$.

First, we consider $g_{V\partial\phi}(\theta, \{\alpha_i\}, \{k_{m,n}\})$ which can be written as:

$$g_{V\partial\phi}(\theta, \{\alpha_i\}, \{k_{m,n}\}) = -\prod_{(m,n)} \sigma_m \times \left( g_{V\partial\phi}^{(1)}(\theta, k_{m,n}) + g_{V\partial\phi}^{(2)}(\{\alpha_i\}, m, k_{m,n}) \right), \quad (A.15)$$

where $\sigma_{m=1,3} = 1$ and $\sigma_{m=2,4} = -1$. The two constituents in Eq. (A.15) are given by

$$g_{V\partial\phi}^{(1)}(\theta, k_{m,n}) := \frac{2\pi i}{(k_{m,n}-1)!} \partial_{z_{m,n}}^{k_{m,n}-1} \left( f(m,n)^{k_{m,n}} \frac{-i\beta\theta}{2\pi z_{m,n}} \right)\Bigg|_{z_{m,n}=y_m}, \quad (A.16)$$

$$
\begin{aligned}
g_{V\partial\phi}^{(2)}(\{\alpha_i\}, m, k_{m,n}) := &\sum_{i\neq m} \frac{2\pi i}{(k_{m,n}-1)!} \partial_{z_{m,n}}^{k_{m,n}-1} \left( f(m,n)^{k_{m,n}} \frac{i\alpha_i'}{y_\mu - z_{m,n}} \right)\Bigg|_{z_{m,n}=y_m} \\
&+ \frac{2\pi i}{k_{m,n}!} \partial_{z_{m,n}}^{k_{m,n}} \left( -i\alpha_m' f(m,n)^{k_{m,n}} \right)\Bigg|_{z_{m,n}=y_m},
\end{aligned}
\quad (A.17)
$$

where

$$f(m,n) = \frac{z_{m,n}^2 - (y_m^2)^*}{z_{m,n} + y_m}. \quad (A.18)$$

At $r = \frac{1}{2}$, the four points around which the contour integrals are performed, are evenly distributed on the unit circle:

$$y_1 = e^{i\frac{\pi}{4}}, \quad y_2 = e^{-i\frac{\pi}{4}}, \quad y_3 = -e^{i\frac{\pi}{4}}, \quad y_4 = -e^{-i\frac{\pi}{4}}. \quad (A.19)$$

Then explicit calculations yield

$$g_{V\partial\phi}^{(1)}(\theta, k_{m,n}) = \begin{cases} 0, & \forall\, k_{m,n}\ \text{even}, \\ \beta\theta, & \forall\, k_{m,n}\ \text{odd}. \end{cases} \quad (A.20)$$

The expression for $g_{V\partial\phi}^{(2)}(\{\alpha_i\}, m, k_{m,n})$ is a bit more complicated. First of all, we have the following relations

$$
\begin{aligned}
g_{V\partial\phi}^{(2)}(\{\alpha_i\}, m=3, k) &= -g_{V\partial\phi}^{(2)}(\{\alpha_i\}, m=1, k), \\
g_{V\partial\phi}^{(2)}(\{\alpha_i\}, m=4, k) &= -g_{V\partial\phi}^{(2)}(\{\alpha_i\}, m=2, k).
\end{aligned}
\quad (A.21)
$$

It then remains to compute $g_{V\partial\phi}^{(2)}(\{\alpha_i\}, m=1, 2, k_{m,n})$, which are given by

$$g_{V\partial\phi}^{(2)}(\{\alpha_i\}, m=1, k_{1,n}) = \begin{cases} 2\pi i c_s(\alpha_2 - \alpha_4), & \text{for } k_{1,n} = 2s+1, \\ 2\pi c_s(\alpha_1 - \alpha_3), & \text{for } k_{1,n} = 2s, \end{cases} \quad (A.22)$$

and

$$g^{(2)}_{V\partial\phi}(\{\alpha_i\}, m=2, k_{2,n}) = \begin{cases} 2\pi i c_s(\alpha_1 - \alpha_3), & \text{for } k_{2,n} = 2s + 1, \\ -2\pi c_s(\alpha_2 - \alpha_4), & \text{for } k_{2,n} = 2s. \end{cases} \tag{A.23}$$

In the above formulae, $c_s$ are rational numbers, with the first few ones (for $k \le 17$) given by

$$\begin{aligned}
c_0 &= \frac{1}{2}, \quad c_1 = \frac{1}{4}, \quad c_2 = \frac{3}{16}, \quad c_3 = \frac{5}{32}, \quad c_4 = \frac{35}{256}, \\
c_5 &= \frac{63}{512}, \quad c_6 = \frac{231}{2048}, \quad c_7 = \frac{429}{4096}, \quad c_8 = \frac{6435}{65536}.
\end{aligned} \tag{A.24}$$

The simplification for $g_{\partial\phi\partial\phi}$ has been worked out in [63].

# B  Details about the $n = 2$ generalized charged moments for chiral level-2 states

We report in this Appendix the contributions to the $n = 2$ charged moments at chiral conformal level-2 due to different combinations of the descendant states in (65) of the vacuum module. For notational convenience, here we will omit the vertex operator charge labeling.

$$\begin{aligned}
F_2^L(\theta; \{1,1\}, \{1,1\}, \{1,1\}, \{1,1\}) &= \frac{\beta^8\theta^8}{1024\pi^8}\sin^8(\pi r) + \frac{\beta^6\theta^6}{512\pi^6}\sin^6(\pi r)(\cos(2\pi r) - 17) \\
&\quad + \frac{\beta^4\theta^4}{4096\pi^4}\sin^4(\pi r)(5\cos(4\pi r) - 84\cos(2\pi r) + 1359) \\
&\quad + \frac{\beta^2\theta^2}{8192\pi^2}\sin^2(\pi r)(3\cos(6\pi r) + 142\cos(4\pi r) - 83\cos(2\pi r) - 8254) \\
&\quad + \frac{1}{131072}(9\cos(8\pi r) + 1080\cos(6\pi r) + 4604\cos(4\pi r) \\
&\qquad\qquad + 37256\cos(2\pi r) + 88123),
\end{aligned}$$

$$\begin{aligned}
F_2^L(\theta; \{1,1\}, \{2\}, \{1,1\}, \{2\}) &= -\beta^6\theta^6\frac{\cos^2(\pi r)\sin^6(\pi r)}{64\pi^6} + \beta^4\theta^4\frac{(25\cos(2\pi r) + 23)\sin^6(\pi r)}{1024\pi^4} \\
&\quad - \beta^2\theta^2\frac{3(7\cos(2\pi r) + 17)\sin^6(\pi r)}{512\pi^2} + \frac{3(9\cos(2\pi r) + 71)\sin^6(\pi r)}{1024},
\end{aligned}$$

$$\begin{aligned}
F_2^L(\theta; \{1,1\}, \{1,1\}, \{2\}, \{2\}) &= -\frac{\beta^6\theta^6}{64\pi^6}\cos^2(\pi r)\sin^6(\pi r) \\
&\quad + \frac{\beta^4\theta^4}{8\pi^4}\cos^6\left(\frac{\pi r}{2}\right)\sin^4\left(\frac{\pi r}{2}\right) \\
&\qquad \times (-2\cos(3\pi r) + 13\cos(2\pi r) + 8\cos(\pi r) + 17) \\
&\quad - \frac{\beta^2\theta^2}{32\pi^2}\cos^6\left(\frac{\pi r}{2}\right)\sin^2\left(\frac{\pi r}{2}\right) \\
&\qquad \times (\cos(4\pi r) - 34\cos(3\pi r) + 64\cos(2\pi r) - 46\cos(\pi r) + 143) \\
&\quad + \frac{1}{128}\cos^6\left(\frac{\pi r}{2}\right) \\
&\qquad \times (25\cos(4\pi r) - 118\cos(3\pi r) + 376\cos(2\pi r) - 778\cos(\pi r) + 623),
\end{aligned}$$

$$\begin{aligned}
F_2^L(\theta; \{1,1\}, \{2\}, \{2\}, \{1,1\}) &= -\frac{\beta^6\theta^6}{64\pi^6}\cos^2(\pi r)\sin^6(\pi r) \\
&\quad + \frac{\beta^4\theta^4}{8\pi^4}\cos^4\left(\frac{\pi r}{2}\right)\sin^6\left(\frac{\pi r}{2}\right) \\
&\qquad \times (2\cos(3\pi r) + 13\cos(2\pi r) - 8\cos(\pi r) + 17) \\
&\quad - \frac{\beta^2\theta^2}{32\pi^2}\cos^2\left(\frac{\pi r}{2}\right)\sin^6\left(\frac{\pi r}{2}\right) \\
&\qquad \times (\cos(4\pi r) + 34\cos(3\pi r) + 64\cos(2\pi r) + 46\cos(\pi r) + 143)
\end{aligned}$$

$$+ \frac{1}{128} \sin^6 \left( \frac{\pi r}{2} \right)$$
$$\times (25 \cos(4\pi r) + 118 \cos(3\pi r) + 376 \cos(2\pi r) + 778 \cos(\pi r) + 623),$$

$$F_2^L(\theta; \{2\}, \{2\}, \{2\}, \{2\}) = \frac{\beta^4 \theta^4}{64 \pi^4} \sin^4(2\pi r) + \frac{\beta^2 \theta^2}{512 \pi^2} \sin^2(2\pi r) (9 \cos(4\pi r) - 4 \cos(2\pi r) - 133)$$
$$+ \frac{1}{32768} (81 \cos(8\pi r) + 56 \cos(6\pi r) + 1628 \cos(4\pi r)$$
$$+ 8072 \cos(2\pi r) + 22931),$$

$$F_2^L(\theta; \{1, 1\}, \{2\}, \{2\}, \{2\}) = -\frac{i \theta^3 \sin^3(\pi r) \cos(\pi r)}{1024 \pi^3} (4 \cos(2\pi r) - 17 \cos(4\pi r) + 141)$$
$$+ \frac{i \theta^5 \sin^5(\pi r) \cos^3(\pi r)}{16 \pi^5}$$
$$- \frac{i \theta \sin^3(\pi r)}{2048 \pi} (754 \cos(\pi r) + 5 \cos(3\pi r) + 9 \cos(5\pi r)),$$

$$F_2^L(\theta; \{1, 1\}, \{1, 1\}, \{1, 1\}, \{2\}) = -\frac{i \theta^7 \sin^7(\pi r) \cos(\pi r)}{256 \pi^7} + \frac{i \theta^5 \sin^5(\pi r)}{1024 \pi^5} (67 \cos(\pi r) - 3 \cos(3\pi r))$$
$$- \frac{i \theta^3 \sin^3(\pi r)}{4096 \pi^3} (650 \cos(\pi r) - 143 \cos(3\pi r) + 5 \cos(5\pi r))$$
$$- \frac{i \theta \sin^3(\pi r)}{4096 \pi} (1466 \cos(\pi r) + 73 \cos(3\pi r) - 3 \cos(5\pi r)).$$

## C  Details about the numerical computations

This part of the appendix is devoted to the description of the numerical tools used to reproduce the data shown in Figs. 4, 5. In particular, we will adapt the computations done in [63] to the evaluation of the generalized normalized charged moments defined in Eq. (4).

We start by introducing the Hamiltonian describing the XX chain of $N$ spins with periodic boundary condition

$$H_{XX} = -\frac{1}{4} \sum_{j=1}^{N} \left( \sigma_j^x \sigma_{j+1}^x + \sigma_j^y \sigma_{j+1}^y \right), \tag{C.1}$$

where $\sigma^\alpha$ are the Pauli matrices. This model can be rewritten in fermionic variables, $\{c_m^\dagger, c_n\} = \delta_{mn}$, after a Jordan-Wigner transformation. The complex fermions $c_n, c_n^\dagger$ can be rewritten in terms of $2N$ Majorana modes as

$$\begin{cases} a_{2m-1} = c_m^\dagger + c_m, \\ a_{2m} = i(c_m^\dagger - c_m), \end{cases} \tag{C.2}$$

and the reduced density matrix can be expressed in terms of the Majorana modes as

$$\rho_A = \frac{1}{Z} e^{1/2 \sum_{jk} B_{jk} a_j a_k}, \tag{C.3}$$

where $B$ is a pure imaginary antisymmetric matrix and $Z$ is the normalization constant. The main goal is then computing the correlation matrix $\Gamma_{mn}^\Omega$ of an eigenstate of the XX spin chain specified by a set of occupied momenta $\Omega = \{k_i\}$. This can be written as

$$\Gamma_{mn}^\Omega = \langle a_m a_n \rangle_\Omega - \delta_{mn}, \tag{C.4}$$

where $\langle \cdots \rangle_\Omega$ means that the expectation value is taken on the state labeled by $\Omega$. It reads

$$\Gamma^\Omega = \begin{pmatrix} \Pi_0 & \Pi_1 & \cdots & \Pi_{N-1} \\ \Pi_{-1} & \Pi_0 & \cdots & \\ \cdots & \cdots & \ddots & \vdots \\ \Pi_{-N+1} & \cdots & & \Pi_0 \end{pmatrix}, \qquad \Pi_m = \begin{pmatrix} g_m^{(1)} & g_m^{(2)} \\ -g_{-m}^{(2)} & g_m^{(1)} \end{pmatrix}, \tag{C.5}$$

where

$$
\begin{cases}
g^{(1)}_{m-n} = \langle a_{2m} a_{2n} \rangle_\Omega - \delta_{mn} = \langle a_{2m-1} a_{2n-1} \rangle_\Omega - \delta_{mn}\,, \\
g^{(2)}_{m-n} = \langle a_{2m-1} a_{2n} \rangle_\Omega\,.
\end{cases}
\tag{C.6}
$$

These quantities can be calculated by relating the correlation functions of Majorana fermions $a_m$ to those of the fermionic variables $c_m$. This is achieved by doing a Fourier transform and reducing the problem to the computation of the correlation functions of free fermions, leading to [67]

$$
\begin{cases}
g^{(1)}_{m-n} = \langle c^\dagger_m c_n \rangle_\Omega + \langle c_m c^\dagger_n \rangle_\Omega - \delta_{mn} = \frac{1}{N}\left[\sum_{k\in\Omega} e^{-i\frac{\pi k}{N}(m-n)} + \sum_{k\notin\Omega} e^{i\frac{\pi k}{N}(m-n)}\right] - \delta_{mn}\,, \\
g^{(2)}_{m-n} = -i\langle c^\dagger_m c_n \rangle_\Omega + i\langle c_m c^\dagger_n \rangle_\Omega = \frac{i}{N}\left[-\sum_{k\in\Omega} e^{-i\frac{\pi k}{N}(m-n)} + \sum_{k\notin\Omega} e^{i\frac{\pi k}{N}(m-n)}\right].
\end{cases}
\tag{C.7}
$$

For a specific bipartition made up of $\ell$ sites, we simply restrict the correlation matrix to a $2\ell$-block of Eq. (C.5), $\Gamma^\Omega_\ell$ [1]:

$$
\Gamma^\Omega_\ell = \begin{pmatrix}
\Pi_0 & \Pi_1 & \cdots & \Pi_{\ell-1} \\
\Pi_{-1} & \Pi_0 & \cdots & \\
\cdots & \cdots & \ddots & \vdots \\
\Pi_{-\ell+1} & \cdots & & \Pi_0
\end{pmatrix}.
\tag{C.8}
$$

In this basis, the charge operator reads

$$
Q_A = \sum_m i a_{2m-1} a_{2m} - i a_{2m} a_{2m-1}.
\tag{C.9}
$$

For a general non-Hermitian matrix $H$, we can use that

$$
\mathrm{Tr}(e^{\frac{1}{2}\sum_{jk} a_j H_{jk} a_k}) = \sqrt{\det(1 + e^H)}\,,
\tag{C.10}
$$

and that the reduced density matrix can be written in terms of the correlation matrix (C.8) as

$$
e^B = \frac{1 + \Gamma^\Omega_\ell}{1 - \Gamma^\Omega_\ell}\,.
\tag{C.11}
$$

Therefore, using the single-particle operators, we find that

$$
\mathrm{Tr}(\rho^n_A e^{i\theta Q_A}) = \sqrt{\det\left(\frac{1 - \Gamma^\Omega_\ell}{2}\right)^n \det\left[1 + \left(\frac{1 + \Gamma^\Omega_\ell}{1 - \Gamma^\Omega_\ell}\right)^n e^{i\theta \tilde{Q}_A}\right]}\,,
\tag{C.12}
$$

where $(\tilde{Q}_A)_{2m,2m-1} = -(\tilde{Q}_A)_{2m-1,2m} = -i$. We stress that this result can be easily extended to different (quadratic) charges and quadratic models, by simply adapting the form of the charge in Eq. (C.9) and computing the new correlators in Eq. (C.7).

The low-lying energy eigenstates in the spin chain can be mapped to the corresponding states in the bosonic CFT described in the main text. For the XX spin chain, this mapping has been detailed in Ref. [67]. When $N$ is an even integer and a multiple of 4, the spin chain states corresponding to the derivative operator and the vertex operator are, respectively:

$$
\begin{aligned}
c_{\frac{N}{4}-\frac{1}{2}} c^\dagger_{\frac{N}{4}+\frac{1}{2}} |0\rangle &\leftrightarrow i\partial\phi\,, \\
c^\dagger_{\frac{N}{4}+\frac{1}{2}} |0\rangle &\leftrightarrow V_{1,0}\,,
\end{aligned}
\tag{C.13}
$$

where $|0\rangle$ is the ground state of the half-filled fermionic model.

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
