# Peer review of "On symmetry-resolved generalized entropies"

_SciPost Physics, doi:SciPost Phys. 18, 211 (2025)_

## Round 1 · Referee Report · Anonymous (Referee 1) · 2025-3-25

Report

In this manuscript, the authors introduce the concept of symmetry-resolved generalized entropies and apply it to systems with a global U(1) symmetry. The aim is to provide a framework for studying symmetry-resolved entanglement in excited states beyond the lowest-energy ones, as well as its evolution in symmetry-preserving out-of-equilibrium settings. The study of symmetry resolution in entanglement measures has gained significant attention in recent years, offering a more refined understanding of the structure of entanglement by decomposing contributions from different symmetry sectors.

Following the standard approach, the authors first define generalized charged moments and then compute them in the (1+1)-dimensional free massless compact boson theory, a model relevant to various physical systems, such as Luttinger liquids. Explicit analytical expressions are provided for the cases n=1 and n=2, cross-checked against known results, and validated through numerical calculations on the XX spin chain. As an application, the authors compute the generalized subsystem charge distribution, which can be used to study the time evolution of full counting statistics. Finally, they derive the symmetry-resolved generalized second Rényi entropy.

Despite being a straighforward extension of previous work, the results contribute to the advancement of this research area, paving the way for the study of symmetry-resolved entanglement in excited states and symmetry-preserving dynamics. The manuscript is well-written, with clear presentation and detailed computations. For these reasons, I support its publication in SciPost.

Recommendation

Publish (easily meets expectations and criteria for this Journal; among top 50%)

---

## Round 1 · Referee Report · Anonymous (Referee 2) · 2025-5-4

Strengths

1- Well-written paper 2- Clear new analytical results

Weaknesses

1- Natural extension of previous work by some of the same authors 2- The relevance to time evolution is not really explored

Report

In this paper, the authors study symmetry-resolved generalized entropies which are relevant to the study of out of equilibrium entanglement in symmetry-preserving setups, or excited states. They perform various exact calculations based on charged moments and replicas on the example of a free compact boson conformal field theory. These results are then checked against lattice fermion computations in the XX chain, with very good agreement.

Overall the manuscript is clearly written, and it contains new results which will be useful to other researcher interested in symmetry resolution in the context of quantum entanglement, or even full counting statistics. For these reasons I recommend publication.

I have two comments:

1) I fail to see the point in the last paragraph in section 1.2. First I would argue that the truncated conformal space approach is already numerical, and the claim that it is 'the only available' method is not really justified in the present manuscript.

2) In appendix C, I find it strange that the authors use Majorana modes instead of complex lattice fermions, since the Hamiltonian is particle-conserving, and the mapping to the bosonic conformal field theory is easier as stated in equation (141). Presumably also, the square-roots would be an artifact of using Majorana modes instead of complex fermions.

Requested changes

I found a typo in page 3: 'an non-abelian' should read 'a non-abelian'.

Recommendation

Publish (meets expectations and criteria for this Journal)

  • validity: high
  • significance: good
  • originality: good
  • clarity: high
  • formatting: excellent
  • grammar: excellent

Author:  Fei Yan  on 2025-05-22  [id 5505]

(in reply to Report 2 on 2025-05-04)

We thank the Referees for very helpful feedback and suggestions. We have made minor modifications accordingly. Below is our response to the two comments of Referee 2.

Comment 1: We thank the Referee for this comment. They are absolutely right that TCSA is already numerical in nature. However, to track the entanglement spreading in generic non-integrable models where an intuitive quasi-particle picture is missing, we would need numerical methods such as TCSA to keep track of the time evolution of the state of the total system in a chosen basis. To our best knowledge, TCSA is the only currently reliable numerical method which allows a direct non-perturbative treatment of out-of-equilibrium quantum field theories. We remark that a promising alternative method to study non-equilibrium quantum field theories has been explored in Ref. [85], though further developments are still needed. Combining TCSA with our analytical results of the symmetry-resolved generalized entropies will then yield a computational framework applicable to track the symmetry resolution of entanglement spreading in generic symmetry-preserving out-of-equilibrium setups. We have added some clarification at the end of Section 1.2 together with a footnote pointing to a more thorough description in Section 2.1 and Ref. [63].

Comment 2: We thank the Referee for this comment. They are perfectly right that the correlation matrix, $\Gamma_{\ell}^{\Omega}$, can be written in terms of complex fermions, and also that the square roots are an artifact of using Majorana modes. This means that it could be written as an $\ell\times\ell$ matrix only defined in terms of $\langle c^{\dagger}_m c_n\rangle$ (which can be read out from our Eq. (135)). However, in the literature about entanglement of excited states, the correlators in terms of Majorana fermions are also frequently used for the XX spin chains (see, for instance, Ref. [67], [78], [63]) and we have simply borrowed the results previously found and adapted them to compute the charged moments. We would prefer to keep using the Majorana description because Eq. (135) can be easily extended to generic quadratic critical spin chains. For instance, if one is interested in doing the symmetry resolution of the parity in the Ising model, by adapting Eq. (135) the final result Eq. (140) still holds. Therefore this would be the most generic formalism to investigate the charged moments, up to modifying the charge and the correlators depending on the specific model one is interested in.

---

## Round 2 · Referee Report · Anonymous (Referee 1) · 2025-5-29

Report

The revised version of the paper can be published

Recommendation

Publish (easily meets expectations and criteria for this Journal; among top 50%)

---

## Round 2 · Referee Report · Anonymous (Referee 2) · 2025-6-1

Report

The authors have convincingly answered my questions, I recommend publication.

Recommendation

Publish (meets expectations and criteria for this Journal)

---

## Editorial Decision

published